# ROBUST CROSS-MODAL SEMI-SUPERVISED FEW SHOT LEARNING

## ABSTRACT

Semi-supervised learning has been successfully applied to few-shot learning (FSL) due to its capability of leveraging the information of limited labeled data and massive unlabeled data. However, in many realistic applications, the query and support sets provided for FSL are potentially noisy or unreadable where the noise exists in both corrupted labels and outliers. Motivated by that, we propose to employ a robust image-text multi-modal semi-supervised few-shot learning (RCFSL) based on Bayesian deep learning. By placing the uncertainty prior on top of the parameters of infinite Gaussian mixture model for noisy input, multi-modality information from image and text data are integrated into a robust heterogenous variational autoencoder. Subsequently, a robust divergence measure is employed to further enhance the robustness, where a novel variational lower bound is derived and optimized to infer the network parameters. Finally, a robust semi-supervised generative adversarial network is employed to generate robust features to compensate data sparsity in few shot learning and a joint optimization is applied for training and inference. Our approach is more parameter-efficient, scalable and adaptable compared to previous approaches. Superior performances over the state-of-the-art approaches on multiple benchmark multi-modal dataset are demonstrated given complicated noise for semi-supervised few-shot learning.

## 1 INTRODUCTION

Despite the impressive success of deep learning models, frequently it requires massive amount of training data to fully demonstrate the potential of the model. In contrast, human is capable of learning new concepts given limited data. Consequently, few-shot learning gathers extensive research interest due to the capabilities of learning new concepts from limited training data. Nevertheless, the success of few-shot learning requires careful handling to robustness and generalization as it is extremely susceptible to noisy labels, outliers as well as adversarial attack Lu et al. (2020a). For instance, in order to automatically recognize several kinds of uncommon animals, only a few annotated images for them are available due to their rarity. Moreover, the images could potentially be corrupted due to an uncontrollable shooting environment or an instrumental malfunction. To mitigate this, one common approach to few shot learning is meta-learning Ren et al. (2018), where the goal is to learn a classifier to distinguish between previously unseen classes, given labeled classes and a larger pool of unseen examples, some of which may belong to the classes of interest, namely semi-supervised few shot learning (SFSL).

Despite the impressive capabilities equipped the ability to leverage unlabeled examples for SFSL, the challenge of lacking novel samples remains to be a bottleneck. Besides visual information, textual data frequently contains rich information and more descriptive concepts for learning. Incorporating image-text multi-modal learning into the framework by training on image-text pairs provides an efficient tool to inject the diversity to the generation process Pahde1 et al. (2021)Pahde1 et al. (2018). The work in Pahde1 et al. (2018) provides a benchmark for multimodal few-shot learning relying on a class-discriminative text conditional generative adversarial network. Later on, Pahde1 et al. (2021) tackles the multimodal few shot learning problem by employing a cross-modal feature generation network to infer the class membership of unseen samples with a simple nearest neighbor approach. Despite of the success of these methods with clean features and perfect labels, the important case that features and labels are contaminated due to out-of-distribution samples, adversarial attack and human fatigue is rarely studied. In parallel, Bayesian deep learning (BDL) has served as a powerful

Figure 1: The overall blockdiagram of the proposed robust cross-modal semi-supervised few shot learning (RCFSL) framework to leverage both the advantage of robust GAN as a high quality generative model and a robust heterogenous semi-supervised VAE as a posterior distribution learner on multi-modality data. Robust multimodal prototypes are then calculated from the robust embedding using the last hidden layer in the discriminator of RCFSL, where three novel components are highlighted in the orange bounding boxes.

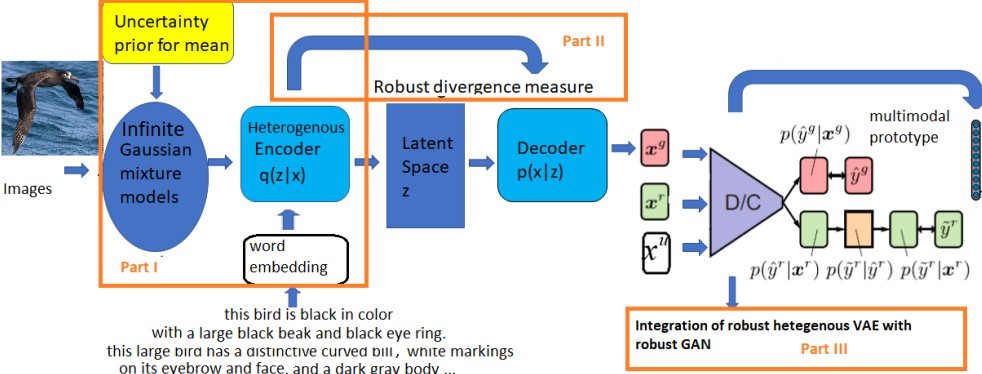

tool in terms of transforming the problem of posterior inference of a BDL model into the optimization of an objective function based on latent variables. Now the question then is: how to design a Bayesian deep learning which counters the noisy labels and outliers jointly in the multimodal semi-supervised few-shot learning. Accordingly, this paper tackles this challenging problem in *robust cross-modal few-shot learning by integrating a deep generative heterogenous model that generalizes well to multi-modality (e.g. image-text modeling) in order to counter noisy labels and outliers.* Specifically, a robust heterogeneous variational auto-encoder is first proposed to encode the noisy visual features and labels in order to jointly learn the information from both modalities by placing the uncertainty prior on the top of the infinite Gaussian mixture models. Subsequently, a robust variational lower bound based on $\beta$-divergence is derived to infer the network parameters. Finally, a robust semi-supervised GAN is integrated with the heterogenous variational auto-encoder by collapsing the generator and the decoder into one to further boost the learning capabilities. RCFSL is built on top of Bayesian deep learning by fusing cross-modal information via the approximation of the joint posterior distributions. In contrast to modality-alignment methods Xing et al. (2019a)Tsai et al. (2017) for robust few-shot learning, the robustness of RCFSL is achieved by accurate modeling of the complicated joint distribution of multi-modality data and robust variational inference by the derived lower bound. Distinct from the work in Xing et al. (2019b) calculating linear combinations in the prototypical representation space, our fusion of multimodality features in the probability distribution is completely data-driven, yielding more robust classification performance in few-shot learning. Major contributions of this paper are: (1) RCFSL harnesses three levels of denoising to ensure the robustness of cross-modal semi-supervised few-shot learning: Firstly, motivated by Hou & Galata (2008), it places the uncertainty prior of the parameters of an infinite Gaussian mixture distribution of image data to avoid mixture components collapsing onto a point or a hyperplane due to outliers. (2) Subsequently, the robust $\beta$-divergence is employed to replace Kullback-Leibler divergence used for data fitting to infer the network parameters and a novel evidence lower bound for semi-supervised few shot learning is derived. (3) Noise-transition layers are applied to both the heterogenous variational encoder and the robust discriminator in semi-supervised learning with an end-to-end training. The performance of RCFSL is further boosted with robust feature generation yielding 7% to 10% absolute accuracy improvement over STOA approaches.

**Related Work** Previous work in multimodal few shot learning frequently tends to first learn text to image mapping to generate additional visual features and then calculate the joint prototype using a weighted average from two representations. Two recent approaches have attracted significant attention in the few-shot learning domain: Matching Networks Vinyals et al. (2016), and Prototypical Networks Snell et al. (2017) where the sample set and the query set are embedded with a neural network, and nearest neighbor classification is exploited relying on a metric in the embedded space. In Oreshkin et al. (2018), metric scaling and metric task conditioning are utilized to improve the performance of few-shot learning algorithms. Kim et al. (2018) and Finn

et al. (2018) employ a probabilistic extension to model-agnostic meta-learning (MAML) framework trained with variational approximation so that the model can generalize well to a new task with a few fine-tuning updates. In Zhang et al. (2020), a bidirectional joint image-text modeling was proposed and VHE-raster-scan-GAN was applied. RCFSL advances the work from Zhang et al. (2020) by improving the robustness of the multi-modal heterogenous encoder and extend the solution to semi-supervised few shot learning. In Tseng et al. (2020), feature-wise transformation layers are utilized for augmenting the image features relying on affine transforms to simulate various feature distributions under different domains for few-shot learning. Different from Tseng et al. (2020), RCFSL augments the robust feature generation from BDL perspective relying on robust semi-supervised GAN. Moreover, our method advances from other robust few shot learning such as Rapnets Lu et al. (2020b) and Adversarial Query Goldblum et al. (2020) by providing mathmatically rigoriously denoising schemes via uncertainty priors and robust divergence in variational inference.

## 2 OUR METHOD

Our approach is to focus on first constructing a semi-supervised robust heterogeneous variational autoencoder leveraging a mixture model to encode both image and text data in few shot learning insensitive to both noisy labels and outliers. Subsequently, a novel robust variational lower bound is derived to facilitate the inference of network parameters relying on $\beta$-divergence for both labeled and unlabeled data. Finally, a robust generative adversairial network is integrated with denoising layers to strengthen the denoising performance togehter with the end-to-end optimization to generate additional visual features to alleviate the sparsity in the semi-supervised few shot learning. Let $\Omega$ denote image space, $\Upsilon$ denote text space and $C = \{1, \ldots, R\}$ be the discrete label space. Further let $x_i \in \Omega$ as the $i$-th input image observation, $\mathbf{t}_i \in \Upsilon$ as its corresponding textual description and $y_i \in C$ as its label. Denote $C_{base}$ as base classes where we have both labeled and unlabeled samples and denote $C_{novel}$ novel classes, which are underrepresented in the data. Inspired by the fact that the student-t distribution is more robust to the outliers than Gaussian distribution by constraining the shapes of the mixture components from collapsing Hou & Galata (2008), we propose to place uncertainty priors (e.g. Gaussian

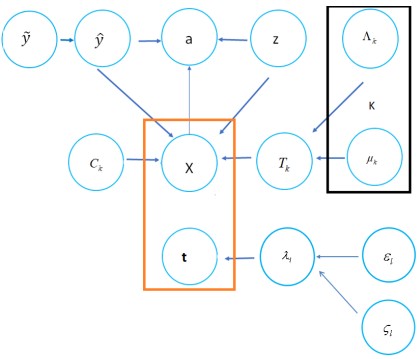

Figure 2: The probabilistic graph model of the proposed RCFSL framework, where $x$ denotes the observations of image data and $\mathbf{t}$ represents the observations of embedded feature vectors for text, $\tilde{y}$ and $\hat{y}$ refer to noisy labels and corrected labels respectively, $z$ and $a$ stand for the latent variable and auxiliary variable for VAE respectively, serving as a bridge to fuse two modalities. $T_k$ and $C_k$ represent the mean and the covariance for the $k$ Gaussian component of image data. $\lambda_l$ represents the parameter in the Poisson distribution for the $l$th feature. Each observation pair $(x_i, \mathbf{t}_i)$ depends on their cluster assignment and distribution parameters and each cluster assignment depends on the stick-breaking procedure $Beta((K-1)\alpha, \chi)$.

priors) on parameters of infinite Gaussian mixture models to *characterize the influence of outliers on image data and constrain the shape of the components to prevent them from collapsing.*

**The Heterogeneous Mixture Model:** Variational autoencoder Diederik & Welling (2013) has been recently proposed as a powerful solution for semi-supervised learning. To the best of our knowledge, this is the first time that a robust heterogenous VAE model has been applied which naturally integrates noisy images and text together which cohesively fuse continuous and discrete multi-modality features. Variational inference are applied to fit the heterogenous model on both of image and text features $\Psi_I$ from base classes $C_{base}$, where the embedding is obtained from the last dense layer right before the softmax output in the discriminator. Once a good mapping $\Psi_I$ based on heterogenous image and text data is learnt, given a test sample, the class membership is given by assigning the class label of the closest prototype to an unseen test sample. In particular, the heterogenous features are first fed into the Dirichlet process mixture and clustered based on their similarity measures, where here image features are modeled as infinite Gaussian mixture distribution Allen et al.

(2019) to characterize the samples from the unseen classes by *computing the posterior distribution for unrepresented clusters more accurately and inferring the number of classes automatically* and text features are characterized by Poisson distribution. Specifically, $x_n$ is a noisy measurement of its true position and is a draw from the Gaussian mixture model, where the mean of each Gaussian component $T_k$ is unknown and the variance $C_k$ is known. In order to characterize the uncertainty and the outliers from the input, the Gaussian prior is placed on the top of the mean for each Gaussian component. Namely, $T_k$ satisfies the normal distribution with the mean $\mu_k$ and the precision matrix $\Lambda_k$. $\omega_i$ is the latent variable for the $i$th data point specifying which Gaussian it came from and $\pi$ is the mixing weight for the Gaussian mixture model. Specifically, a Normal-Wishart prior Murphy (2007) is placed on the mean and precision of the Gaussian components: $p(\mu, \Lambda) = \prod_{k=1}^{K} N(\mu_k|m_0, (\beta_0 \Lambda_k)^{-1})W(\Lambda_k|W_0, \nu_0)$ , where $m_0$ is set to be zero and $\beta_0$ is set to be a very small value. $W(\Lambda|W, \nu)$ is a Wishart distribution with scale matrix $W$ and $\nu$ degrees of freedom. For text features after word embeddings, $\epsilon_l$ and $\zeta_l$ are the prior shape and rate parameters respectively in the Gamma distribution that generates the average rate parameter for the $l$-th Poisson feature. In particular, the cluster assignments for each observation are drawn from multinomial distributions where the prior parameters represent the mixing weights of the corresponding clusters. Specifically, the truncated stick-breaking process is employed to construct mixing weights. By construction, a single parameter $\alpha$ controls the portion of the $K-1$ major clusters, and $\chi$ separately controls for the portion of the remainder via a stick-breaking procedure $Beta((K-1)\alpha, \chi)$ Blei & Jordan (2017). Namely, where $\omega = [\omega_1, \ldots, \omega_K]$ belongs to the $(K-1)$-dimensional simplex and is generated from a Dirichlet prior with two parameters $p(\omega)\tilde{D}ir(\alpha, \ldots, \alpha, \chi)$. Typically, given the image data $x$ and the textual features $\mathbf{t}$, variational inference with deep learning from powerful probabilistic models are constructed by an inference neural network $q(z|x, \mathbf{t})$ and a generative neural network $p(x, \mathbf{t}|z)$. The generative model and the inference model are parameterized by $\theta$ and $\phi$ respectively. Subsequently, the network parameters are estimated by optimizing the evidence lower bound (ELBO) in the variational inference. For unsupervised learning of multimodality data, the vanilla VAEs are optimized by maximizing ELBO:

$$ELBO = \mathrm{E}_{p_{data}(\mathbf{t}, x)}[L(\mathbf{t}, x)], L(x) := \mathrm{E}_{z \sim q(z|x, \mathbf{t})}[\ln p(\mathbf{t}|z)] - \mathrm{D}_{KL}[q(z|x, \mathbf{t})//p(z)] \qquad (1)$$

**The Generative and Inference Model:** We shall now present the proposed generative and inference model for few-shot learning. Given a semi-supervised learning setting, the labels $y$ are either unknown (for unlabeled data) or noisy (for labeled data). The generative model is then defined as: $p_\theta(x, \mathbf{t}|\omega, T, C)p_\theta(T|Z, \mu, \Lambda, \lambda)p_\theta(a|z, \tilde{y}, x, \mathbf{t})p_\theta(x, \mathbf{t}|\tilde{y}, z)p(z)p(\tilde{y})p(\omega)p(\mathbf{t}|\lambda)$. Define $p_\theta$ as the deep neural network with parameters $\theta$ and $y$ as the ground truth of class label. For unlabeled data, $y$ is considered as a latent variable. Further denote $\mathrm{Cat}(.)$ as a multinomial distribution and in this paper we reasonably assume that the labels are of categorical distribution and the proposed model applies to other distributions for the latent variable $y$. In order to fit more complicated posteriors for the marginal distribution $p(z|x, \mathbf{t})$ and bridge the gap between two modalities, motivated by Maaløe et al. (2017), we extend the variational distribution with auxiliary variables $a$, so that the generative model is invariant to marginalization over $a$ $p(x, z, a, \omega, T, C, \mu, \Lambda) = p_\theta(x|\omega, T, C)p_\theta(T|Z, \mu, \Lambda)p_\theta(a|z, x)p_\theta(x, z)$. To attenuate the influence with the noisy labels, further denote $\tilde{y}$ as the corrupted labels and $\hat{y}$ as the corrected label after denoising layer. Define $K \times K$ noise transition matrix $M$ to associate $\tilde{y}$ with $\hat{y}$ and estimation of $M$ has been addressed in previous methods Sukhbaatar et al. (2015)Patrini et al. (2017). In particular, $M = (M_{i,j}) \in [0,1]^{c \times c}(\sum_i M_{i,j} = 1)$. The proposed generative model can then be expressed as:

$$p(z) = N(z|0, I) , p(\tilde{y}) = \mathrm{Cat}(\tilde{y}|\eta) , p_\theta(a|z, \tilde{y}, x, \mathbf{t}) = f(a; z, \tilde{y}, x, \mathbf{t}) ,$$

$$p_\theta(x, \mathbf{t}|z, \tilde{y}) = f(x, \mathbf{t}; z, \tilde{y}, \theta) , p(T_k|\mu_k, \Lambda_k) = N(t_k|\mu_k, \Lambda_k^{-1}), p(\hat{y} = i|\tilde{y} = j) = M_{ij} , \qquad (2)$$

$$\begin{cases} p(x, \mathbf{t}|T, C, \lambda) = N(x_n|t_k, C_k) \prod_{n=1}^{N_l} N(t_n|t_{\tilde{y}_n}, C_{\tilde{y}_n})\frac{\lambda^k \exp^{-\lambda}}{k!} \\ \text{if the } j\text{th cluster is represented} \\ p(x, \mathbf{t}|T, C, \lambda) = \int p(x_n|t_k, C_k)p(t_k|\mu_k, \Lambda_k)p(\mu_k, \Lambda_k)d\mu_k d\Lambda_k \frac{\lambda^k \exp^{-\lambda}}{k!} \\ \text{if the } j\text{th cluster is unrepresented} \end{cases}$$

The inference model can be represented as:

$$q_\phi(a, z, \mu, \Lambda, T, \tilde{y}, \lambda|x, \mathbf{t}) = q(z|a, \tilde{y}, x, \mathbf{t})q(a|x, \mathbf{t})q(\tilde{y}|a, x, \mathbf{t})q(T, \mu, \Lambda, \lambda|x, \mathbf{t})$$

$$q_\phi(z|a, \tilde{y}, x, \mathbf{t}) = N(z|\mu_\phi(a, \tilde{y}, x, \mathbf{t}), diag(\sigma^2)), q_\phi(\tilde{y}|a, x, \mathbf{t}) = \mathrm{Cat}(\tilde{y}|\eta_\phi(a, x, \mathbf{t})),$$

$$q_\phi(\mu_k, \Lambda_k) = q(\mu_k|\Lambda_k)q(\Lambda_k), q(\lambda_l|\mathbf{t}) \propto Gamma(\lambda_l|\epsilon_l + \textstyle\sum_i \psi_{ik}\mathbf{t}_{il}, \zeta_l + \textstyle\sum_i \psi_{ik}) , \qquad (3)$$

where $q(\lambda_l|\mathbf{t})$ characterizes posterior variational density of Poisson parameters given the discrete embedded feature vectors $\mathbf{t}$ converted from text data. $\psi_{ik}$ stands for the probability of the $i$th observation belongs to the cluster $k$. $t_{il}$ represents the $l$-th Poisson feature of the $i$th observation. Denote $N_{-n,j}$ as the number of data points, excluding $x_n$ which belongs to the mixture component $j$. To compute $q(T, \mu, \Lambda, \lambda|x, \mathbf{t})$, mean-field approximation is applied Bishop (2006) to factorize all the latent variables and parameters for the represented and unrepresented $j$th cluster respectively:

$$q(T, \mu, \Lambda, \lambda|x, \mathbf{t}) = \frac{N_{-n,j}}{N-1+\alpha} N(x_n|t_k, C_k) \prod_{n=1}^{N_l} N(t_n|t_{\tilde{y}_n}, C_{\tilde{y}_n}) q(\lambda_l|\mathbf{t})$$

$$q(T, \mu, \Lambda, \lambda|x, \mathbf{t}) = \frac{\alpha}{N-1+\alpha} \int p(x_n|t_k, C_k) p(t_k|\mu_k, \Lambda_k) p(\mu_k, \Lambda_k) d\mu_k d\Lambda_k q(\lambda_l|\mathbf{t}) \quad (4)$$

The above equation indicates that given the multimodality data, the unlabeled samples has a certain probability of being classified as unrepresented mixtures (e.g. unseen new classes), which facilitates the learning capabilities of the proposed robust semi-supervised few shot learning. Within an iteration, some unlabeled samples are associated with unrepresented mixtures, a new represented mixtures will emerge which successfully addressed the challenges in semi-supervised few shot learning when the unlabeled data contains unseen new classes which doesnot exist in labeled data.

**Robust Variational Lower Bound:** To further alleviate the impact from outliers, the robust divergence is employed to infer network parameters more accurately. The theoretical foundation of $\beta$-divergence has been initially defined at Basu et al. (1998), where the $\beta$-divergence between two functions $g$ and $f$ are defined as

$$D_\beta(g \| f) = \frac{1}{\beta} \int g(x)^{1+\beta} dx + \int f(x)^{1+\beta} dx - \frac{\beta+1}{\beta} \int g(x) f(x)^\beta dx \quad (5)$$

When $\beta \to 0$, the $\beta$-divergence converges to KL-divergence, $\lim_{\beta \to 0} D_\beta(g \| f) = D_{KL}(g \| f)$. As described in Futami et al. (2018), minimizing the $\beta$-divergence from the empirical distribution $\hat{p}(x)$ to $p(x; \theta)$ $\arg\min_\theta D_\beta(\hat{p}(x) \| p(x; \theta))$, it is easy to show $\frac{1}{N} \sum_{i=1}^N p(x_i; \theta)^\beta \frac{\partial \ln p(x_i; \theta)}{\partial \theta} - E_{p(x;\theta)}[p(x; \theta)^\beta \frac{\partial \ln p(x; \theta)}{\partial \theta}]$. As the probability densities of outliers are usually much smaller than those of inliers, the first term of the above equation is the likelihood weighted according to the power of the probability density for each sample, which effectively suppress the likelihood of outliers. This estimator is also called as $M$-estimator Huber & Ronchetti (2011), which provides provably superior performance in various machine applications Li & Gal (2017). The variational lower bound for the proposed RCFSL model for labeled data can be represented as

$$\log p(x, \tilde{y}, \mathbf{t}) = \int_a \int_z \int_T \int_\mu \int_\Lambda \int_\lambda \int_\omega \log(x, \tilde{y}, a, z, T, \mu, \Lambda, \lambda, \mathbf{t}) da dz dT d\mu d\Lambda d\lambda d\omega \geq$$

$$E[\log(p_\theta(a, z, T, \mu, \Lambda, \lambda, \omega, x, \tilde{y}, \mathbf{t}))] - E[q_{\phi(a,z,T,\mu,\Lambda,\omega|x,\mathbf{t},\tilde{y})}] = E[\log(p_\theta(a, z, T, \mu, \Lambda, \omega, x, \mathbf{t}, \tilde{y}))]$$

$$- E[q_{\phi(a|x,\mathbf{t})}] - E[q_{\phi(z|a,\tilde{y},x,\mathbf{t})}] - E[q_{\phi(T|\mu,\Lambda,x)}] - E[q_{\phi(\mu,\Lambda)}] - E[q_{\phi(\omega|\alpha)}] - E[q_{\phi(\lambda|\mathbf{t})}]$$

The above inequality can be rewritten as

$$\log p(x, \tilde{y}, \mathbf{t}) \geq E_{q_{\phi(a,z,T,\mu,\Lambda,\lambda,\omega|x,y,\mathbf{t})}}[\log \frac{p_\theta(a,z,T,\mu,\Lambda,\lambda,x,\tilde{y},\mathbf{t})}{q_{\phi(a,z,T,\mu,\Lambda,\lambda,\omega|x,\mathbf{t},\tilde{y})}}] = E_{q_\phi(a,z,T,\mu,\Lambda,\lambda,\omega|x,\mathbf{t},\tilde{y})}$$

$$[\log(p_\theta(x, \mathbf{t}, \tilde{y}|a, z, T, \mu, \Lambda, \lambda, \omega))] + D_{KL}[q(a, z, T, \mu, \Lambda, \lambda|x, \mathbf{t}, \tilde{y}) // p(a, z, T, \mu, \Lambda, \lambda)] \quad (6)$$

To mitigate the influence of outliers, let $H = \{a, z, T, \omega, \mu, \Lambda, \lambda\}$ represent the set of all the latent variables and leverage the technique from Futami et al. (2018), we can replace KL-divergence with $\beta$-Divergence and cast the $\beta$-ELBO for labeled data $L_\beta$ as:

$$L_\beta = \int q(H|x, \mathbf{t}, \tilde{y})(-\frac{\beta+1}{\beta} \sum_{i=1}^N p(\tilde{y}_i|H; x_i, \mathbf{t}_i)^\beta + N \int p(\tilde{y}|H; x, \mathbf{t})^{1+\beta} d\tilde{y}) + D_{KL}[q(H|x, \mathbf{t}, \tilde{y}) // p(H)] \quad (7)$$

For unlabeled data, by introducing the variational distribution for $\tilde{y}$ as $q_\phi(a, x, \mathbf{t}|\tilde{y})$, the variational lower bound for the proposed RCFSL can be represented as

$$\log p(x, \mathbf{t}) = \int_a \int_z \int_T \int_\mu \int_\Lambda \int_{\tilde{y}} \log(x, \tilde{y}, a, z, T, \mu, \Lambda) da dz dT d\mu \Lambda d\tilde{y} \geq E_{q_\phi(a,\tilde{y},z,T,\mu,\Lambda|x,\mathbf{t})}$$

$$[\log \frac{p_\theta(a,z,T,\mu,\Lambda,x,\mathbf{t},\tilde{y})}{q_\phi(a,z,T,\mu,\Lambda,\lambda,\tilde{y}|x,\mathbf{t})}] = E[\log(p_\theta(a, z, T, \mu, \Lambda, \omega, \lambda, x, \mathbf{t}, \tilde{y}))] - E[q_{\phi(a|x,\mathbf{t})}] - E[q_{\phi(\tilde{y}|a,x,\mathbf{t})}]$$

$$- E[q_{\phi(z|a,\tilde{y},x,\mathbf{t})}] - E[q_{\phi(T|\mu,\Lambda,x)}] - E[q_{\phi(\mu,\Lambda)}] - E[q_{\phi(\omega|\pi)}] - E[q_{\phi(\lambda|\mathbf{t})}]$$

Similarly, replacing the KL-divergence with $\beta$-Divergence and augmenting the latent variable $H_u = \{a, z, y, T, \omega, \mu, \Lambda, \lambda\}$, the $\beta$-ELBO for unlabeled data in RCFSL is:

$$U_\beta = \int q(H_u|x, \mathbf{t})(-\frac{\beta+1}{\beta} \sum_{i=1}^N p(x_i, \mathbf{t}_i|H_u)^\beta + \int p(x, \mathbf{t}|H_u)^{1+\beta} dx) + D_{KL}[q(H_u|x, \mathbf{t}) // p(H_u)],$$

Practically, $L_\beta$ and $U_\beta$ are calculated via Monte Carlo sampling. The robustness of our proposed ELBO can be guaranteed leveraging the influence function (IF) Futami et al. (2018)Huber & Ronchetti (2011). As IF is widely used to analyze how much contamination affects estimated statistics, it is straightforward to show that given the perturbation on the empirical cumulative distribution caused by outliers, it is straightforward to show that the IF of our posterior distribution is bounded. The objective function for robust cross-modal variational autoencoder is: $L_{RCVAE} = L_\beta + \lambda_1 U_\beta$ , where $\lambda_1$ represents the weight to control the trade-off between the labeled data and unlabeled data.

**Robust Feature Generation:** The construction of the proposed uncertainty priors and the robust divergence measure in our framework aims at better approximation of the posterior distribution under noisy labels and outliers. This is also related to employing generative adversarial learning of $z$ and $x, \mathbf{t}$ defined by the data, the encoder, the prior and decoder. Different from VAEs that assumes parametric data distribution and perform posterior inference, GANs in general utilize implicit data distribution and do not provide meaningful latent representations. By learning both a generator $G$ and a discriminator $D$, a min-max objective is optimized:

$$\min_G \max_D \{E_{x,\mathbf{t}\sim p_{data}(x,\mathbf{t})}[\ln(D(x,\mathbf{t}))] + E_{z\sim p(z)}[\ln(1 - D(G(z)))]\} \tag{8}$$

In Kaneko et al. (2019), a noise transition model is incorporated to learn a clean label conditional generative distribution. But their model only considered noisy labels without outliers and is limited to supervised learning. Recent work on semi-supervised GAN with $k$ classes Kumar et al. (2017)Salimans et al. (2016) modify the discriminator with $k + 1$ outputs on the discriminator by considering the fake images as the $k + 1$th class. Hence, the loss for the training can be computed with a supervised loss and an unsupervised loss respectively:

$$L = L_{sup} + L_{unsup} = -E_{x,\mathbf{t},y\sim p_d(x,\mathbf{t},y)} \log p_f(y|x,\mathbf{t}, y \le k) -$$

$$E_{x,\mathbf{t}\sim p_g(x,\mathbf{t})} \log(p_f(y = k+1|x,\mathbf{t})) - E_{x,\mathbf{t}\sim p_d(x,\mathbf{t})} \log(1 - p_f(y = k+1|x,\mathbf{t})) , \tag{9}$$

where $L_{sup}$ represents the negative log probability of the label given the data is real. Denote $D(x,\mathbf{t}) = 1 - p_f(y = k+1|x,\mathbf{t})$, the loss for semi-supervised GAN on multimodality data $L_{sGAN}$ can be written as: $L = -E_{x,\mathbf{t}\sim p_d(x,\mathbf{t})} \log D(x,\mathbf{t}) - E_{z\sim noise}(1 - D(G(z))) - E_{x,\mathbf{t},y\sim p_d(x,\mathbf{t},y)} \log p_f(y|x,\mathbf{t}, y \le k)$ . More recently, the label noise robust GAN (rGAN) Kaneko et al. (2019) has achieved promising performance in classifying images with noisy labels by incorporating a noise transition model to learn a clean label conditional generative distribution under noisy labels. Thus, given the noise samples $(\tilde{x}^r, \tilde{y}^r) \sim \tilde{p}_d(x, \tilde{y})$, to construct a label-noise robust conditional generator and discriminator, the objective function of the robust semi-supervised GAN in the proposed robust cross-modal semi-supervised few-shot learning is expressed as: $L_{RSGAN} = -E_{x,\mathbf{t}\sim p_d(x,\mathbf{t})} \log(1 - \tilde{C}(\tilde{y} = k+1|x,\mathbf{t})) - E_{z\sim noise}(1 - D(G(z))) - E_{(\tilde{x}^r,\mathbf{t},\tilde{y}^r)\sim \tilde{p}_d(x,\mathbf{t},\tilde{y})} \log \tilde{C}(\tilde{y} = \tilde{y}^r|x,\mathbf{t},\tilde{y} \le k) = -E_{x,\mathbf{t}\sim p_d(x,\mathbf{t})} \log(1 - M_{\hat{y}^r,\tilde{y}^r}\hat{C}(\hat{y} = k+1|x,\mathbf{t})) - E_{z\sim noise}(1 - D(G(z))) - E_{(\tilde{x}^r,\mathbf{t},\tilde{y}^r)\sim \tilde{p}_d(x,\mathbf{t},\tilde{y})} \log \sum_{\hat{y}^r} M_{\hat{y}^r,\tilde{y}^r}\hat{C}(\hat{y} = \hat{y}^r|x,\mathbf{t},\hat{y} \le k)$ . Without modeling the data distribution explicitly and representing the latent space in a meaningful manner, it does not provide the functionality to counter the outliers in the generator and discriminator. Motivated by the thought of bridging the gap between robust VAE and robust GAN, we feed the robust variational posterior $p(z|x,\mathbf{t})$ instead of the random noise $p(z)$ into the label noise robust semi-supervised GAN as the source of randomness as both of the decoder and the generator of RSGAN share the mapping from $z$ to $x$ and $\mathbf{t}$. The full optimization function for the proposed RCFSL framework can be represented as:

$$\min_{G_{RCFSL}} \max_D E_{p_{data}(x,\mathbf{t})}[L(x,\mathbf{t})]L(x,\mathbf{t}) := \ln D(x,\mathbf{t}) + D_{KL}[q(H|x,\mathbf{t},\tilde{y})//p(H)] +$$

$$\lambda_1 D_{KL}[q(H_u|x,\mathbf{t})//p(H_u)] + E_{z\sim q(H|x,\mathbf{t},\tilde{y})}[\ln(1 - D(G_{RSGAN}(z)) + (-\frac{\beta+1}{\beta}\sum_{i=1}^N p(\tilde{y}_i|H; x_i,\mathbf{t}_i)^\beta +$$

$$N \int p(\tilde{y}|H; x,\mathbf{t})^{1+\beta}d\tilde{y} + \lambda_1 E_{z\sim q(H_u|x,\mathbf{t})}[(-\frac{\beta+1}{\beta}\sum_{i=1}^N p(x_i,\mathbf{t}_i|H_u)^\beta + \lambda_1 N \int p(x,\mathbf{t}|H_u)^{1+\beta}dx)] , \tag{10}$$

where $\lambda_1$ is set to be the ratio of unlabled samples verus labeled samples and the discriminator loss is characterized by $E[\ln D(x,\mathbf{t})] = -E_{x,\mathbf{t}\sim p_d(x,\mathbf{t})} \log(1 - M_{\hat{y}^r,\tilde{y}^r}\hat{C}(\hat{y} = k+1|x,\mathbf{t})) - E_{(\tilde{x}^r,\mathbf{t},\tilde{y}^r)\sim \tilde{p}_d(x,\mathbf{t},\tilde{y})} \log \sum_{\hat{y}^r} M_{\hat{y}^r,\tilde{y}^r}\hat{C}(\hat{y} = \hat{y}^r|x,\mathbf{t},\hat{y} \le k)$ . This new architecture is expected to better predict the class labels under the compound noise. We then train RCFSL including the robust heterogenous encoder, the generator and the robust discriminator in an end-to-end manner using adaptive moment estimation (Adam) Kingma & Ba (2015). Each multimodal embedding prototype $\mathbf{p}_c$ (of category $c$) is computed by averaging the embeddings of all support samples of class $c$. Once the robust embedding is obtained, the distance between the embedding of the query $q_t$ and the multimodal prototype $\mathbf{p}_c$ is calculated by $p(y = c|\mathbf{p}_c) = \frac{\exp(-d(f(q_t),\mathbf{p}_c))}{\sum_k \exp(-d(f(q_t),\mathbf{p}_k))}$ where $d$ refers to Euclidean distance and the query is classified as the class with the minimum distance.

Table 1: Semi-supervised few-shot classification accuracy on test split of miniImageNet. The top row of uni-modality results are only applied to visual features. The middle row reports the classification performance on the methods of cross modality alignment based methods extended to few shot learning framework and the last row demonstrates the results with robust few shot learning.

| Dataset | mIN(5N 3K 1C) | mIN(5N 5K 2C) | mIN(5N 10K 3C) |
|---|---|---|---|
| Uni-modality few shot learning methods | | | |
| MAMLFinn et al. (2017) | $43.35\pm 0.81\%$ | $51.64\pm 0.47\%$ | $- - -$ |
| Matching NetworkVinyals et al. (2016) | $40.23\pm 0.75\%$ | $49.68\pm 0.63\%$ | $- - -$ |
| Prototypical NetworkSnell et al. (2017) | $46.72\pm 0.59\%$ | $63.49\pm 0.36\%$ | $71.27\pm 0.59\%$ |
| Discriminative k-shotBauer et al. (2017) | $51.64\pm 0.37\%$ | $68.36\pm 0.42\%$ | $72.83\pm 0.25\%$ |
| SNAILMishra et al. (2018) | $53.22\pm 0.28\%$ | $69.16\pm 0.48\%$ | $73.45\pm 0.13\%$ |
| CAMLJiang et al. (2019) | $57.87\pm 0.62\%$ | $70.67\pm 0.39\%$ | $- - -$ |
| Cross modality modeling extended to metric-based FSL framework | | | |
| CBPL-FSLXing et al. (2019a) | $56.39\pm 0.27\%$ | $73.11\pm 0.45\%$ | $- - -$ |
| ReViSE-FSLTsai et al. (2017) | $41.69\pm 0.69\%$ | $64.25\pm 0.34\%$ | $72.35\pm 0.47\%$ |
| f-CLSWGAN-FSLXian et al. (2018) | $51.63\pm0.76\%$ | $53.62\pm 0.26\%$ | $73.71\pm 0.59\%$ |
| CADA-VAE-FSLSchönfeld et al. (2019) | $56.33\pm0.45\%$ | $71.97\pm0.27\%$ | $75.42\pm 0.60\%$ |
| AM3+TADAMXing et al. (2019b) | $64.31\pm 0.37\%$ | $74.03\pm0.55\%$ | $77.60\pm 0.49\%$ |
| VHE-raster-scan-GANZhang et al. (2020) | $78.65\pm 0.42\%$ | $79.23\pm0.75\%$ | $78.52\pm 0.35\%$ |
| Robust few shot learning methods | | | |
| AQGoldblum et al. (2020) | $73.42\pm 0.63\%$ | $76.26\pm 0.58\%$ | $79.05\pm 0.42\%$ |
| RAPNetLu et al. (2020b) | $75.72\pm 0.51\%$ | $79.87\pm 0.47\%$ | $81.24\pm 0.64\%$ |
| RCFSL(ours) | $\mathbf{84.63\pm 0.35\%}$ | $\mathbf{85.35\pm 0.39\%}$ | $\mathbf{87.21\pm 0.39\%}$ |

## 3 EXPERIMENTS

**Dataset and Competing Methods:** We extensively evaluate the proposed robust cross-modal semi-supervised few shot learning (RCFSL) algorithm on multiple benchmark datasets including mini-ImageNet(mIN), tieredImageNet (tIN), Fewshot-CIFAR100 (FC100) and Caltech-UCSD Birds 200-2011 (CUB). The miniImageNet dataset is a subset of ILSCRS-12 which includes 100 classes where each class contains 600 RGB images with size 84 by 84. We split the datasets by using 64 classes for training, 16 classes for validation and 20 classes for testing where 16 classes are utilized to monitor the model's generalization performance. The CUB Bird dataset contains 11788 images of 200 different bird species, where the data is split equally in training and testing data. Namely, for each category, 30 training image and 30 testing images are included, where in the training dataset, 10% images are labeled images and the rest of the images are unlabeled. 10 short textual descriptions per image are provided by Reed et al. (2016). Following the work Zhang et al. (2017), the text encoder pretrained by Reed et al. (2016) is employed and the data is split such as $|C_{Base}| = 150$, $|C_{Novel}| = 50$. Specifically, $n = \{1, 2, 5, 10, 15, 20\}$ images are selected from novel classes in order to perform few shot learning. We evaluate the two types of label noise including symmetric noise and asymmetric noise. For symmetric noise, we inject the label noise by randomly flipping the labels of the labeled data into a different label in the novel classes. For asymmetric noise, the corrupted labels are generated by replacing the correct labels with their most similar classes (using nearest neighbor measurement). Outliers are mimicked by including samples from data distributions far away from the training data distributions, namely out-of-distribution (OOD) samples. GloVe Pennington et al. (2014) is utilized to extract the word embeddings for the category labels, where the embeddings are trained with large unsupervised text corpora. RCFSL is compared with multiple benchmark methods and state-of-the-art approaches. Typically, similarly as Xing et al. (2019b), three families of methods are evaluated: (1) uni-modality few-shot learning methods such as MAMLFinn et al. (2017), LEORusu et al. (2019) and CAMLJiang et al. (2019). In particular, latent embedding optimization (LEO) Rusu et al. (2019) addresses MAML problem by relying on a few updates on a low data regime to train models in a high dimensional parameter space. (2) cross modality few shot learning methods ReViSE Tsai et al. (2017), CADA-VAE Schönfeld et al. (2019) Xing et al. (2019b), VHE-raster-scan-GANZhang et al. (2020) and Feature transformTseng et al. (2020). Among them, ReViSE Tsai et al. (2017) minimizes maximum mean discrepancy (MMD) of the distributions from two representation spaces for better alignment. CADA-VAE Schönfeld et al.

Table 2: Semi-supervised few-shot classification accuracy on test split of tieredImageNet (tIN). The top row of uni-modality results are only applied to visual features. The middle row reports the classification performance on the methods of cross modality alignment based methods extended to few shot learning and the last row demonstrates the results with robust few shot learning.

| Dataset | tIN(5N 3K 1C) | tIN(5N 5K 2C) | tIN(5N 10K 3C) |
|---|---|---|---|
| Uni-modality few shot learning methods | | | |
| MAMLFinn et al. (2017) | 47.65± 0.66% | 52.63± 0.71% | − − − |
| Prototypical NetworkSnell et al. (2017) | 49.65± 0.56% | 68.13± 0.57% | 73.57 ± 0.67% |
| Discriminative k-shotBauer et al. (2017) | 51.64± 0.37% | 69.27± 0.42% | 74.06 ± 0.25% |
| SNAILMishra et al. (2018) | 55.63± 0.31% | 70.23± 0.48% | 75.69 ± 0.43% |
| CAMLJiang et al. (2019) | 58.76± 0.45% | 71.35± 0.32% | − − − |
| Cross modality modeling extended to metric-based FSL framework | | | |
| CBPL-FSLXing et al. (2019a) | 58.57± 0.45% | 74.35 ± 0.67% | − − − |
| ReViSE-FSLTsai et al. (2017) | 44.53± 0.81% | 67.61± 0.85% | 73.65± 0.56% |
| Feature transformTseng et al. (2020) | 59.78±0.85% | 73.21± 0.42% | 75.23± 0.63% |
| CADA-VAE-FSLSchönfeld et al. (2019) | 57.64±0.59% | 71.97±0.27% | 77.31± 0.57% |
| AM3+TADAMXing et al. (2019b) | 65.92± 0.53% | 74.03±0.55% | 78.34± 0.45% |
| VHE-raster-scan-GANZhang et al. (2020) | 73.41± 0.63% | 76.35±0.75% | 79.03± 0.32% |
| Robust few shot learning methods | | | |
| AQGoldblum et al. (2020) | 75.31± 0.56% | 78.35± 0.62% | 81.33± 0.48% |
| RAPNetLu et al. (2020b) | 76.48± 0.51% | 80.11± 0.53% | 82.61± 0.39% |
| RCFSL(ours) | **85.65± 0.47%** | **87.63± 0.34%** | **88.57± 0.42%** |

(2019) leverages two VAEs to embed information for both modalities and align the distribution of the two latent spaces. (3) Robust few shot learning methods including robust attentive profile networks (Rapnets) Lu et al. (2020b) and Adversarial Querying (AQ) Goldblum et al. (2020). RapnetsLu et al. (2020b) consists of an embedding module, a correlation module, and an attentive module to suppress the outliers in few-shot learning. Adversarial Querying Goldblum et al. (2020) produces adversarially robust meta-learners and achieves efficient classification for few-shot learning tasks.

**Implementation Details:** For miniImagenet, tiredImagenet and FC100 datasets, we parameterize the heterogeneous VAE with three sets of 5 by 5 fully convolutional, ReLU and pooling layers followed by two fully connected hidden layers where each pair of layers contains the hidden units as dim(h) = 1000. We further set the dimension of the auxiliary variable $a$ and the latent variable $z$ to be 200 respectively. For the CUB dataset, we leverage Resnet-18 as backbone and incorporate the uncertainty prior for the input, the noise transition model along with the auxiliary variables as our encoder. The network is trained with SGD using a batch size of 128. A momentum of 0.9 is set with a weight decay of 0.0005. The network is trained for 30000 iterations. The initial learning rate is set as 0.1, and reduce it by a factor of 10 after iterations 15,000, 17,500 and 19,000. For the configurations of discriminator in GAN, we follow the same configuration of DC-GAN and follow the similar discriminator Mirza & Osindero (2014) for DC-GAN. $\beta$ is varied from 0.1 to 0.4 where the best performance is reported. The similar setup in Xing et al. (2019b) is utilized relying on $ACC(N, K, C)$ to evaluate the performance of classification accuracy in a $N$-way and $K$-shot SFSL settings, where $C$ represents the total number of outliers and noisy labels in each class. We use 10 textual descriptions per image to generate multimodal prototypes. The semi-supervised few-shot classification accuracy on test split of miniImageNet is shown in Table 1. As shown in Table 1, RCFSL achieves the best classification accuracy compared to all of the competing methods in different cases. The large performance margin suggests the proposed mechanism not only fuses two modalities of features cohesively and but also effectively suppresses the noisy labels and outliers in semi-supervised few-shot learning. Compared to the adaptive modality mixture mechanism (AM3) with TADAM as backbone Xing et al. (2019b), RCFSL has the advantage of fusing multimodal distributions in the heterogeneous encoder via the latent space to fully utilize the multimodal feature representation instead of computing multimodal prototypes with a convex combination afterwards. In contrast to robust FSL methods like Rapnets Lu et al. (2020b) and AQ Goldblum et al. (2020), the performance gain can be mainly attributed to the fact that RCFSL employs a mathematically rigorous variational inference insensitive to noisy labels and outliers to optimize network parameters, yielding more accurate classification performance. The margin in performance is particularly

Table 3: Semi-supervised few-shot learning results (%) using our method with different combinations of denoising strategies training with different levels of label noise and outliers. The bold number in each column of sub-boxes represents the best result. Here UP stands for uncertainty priors, RD means robust divergence and CM represents cross multimodal data (images and text).

| Dataset | Denoising Strategies | | | mIN | tIN | FC100 | CUB-200 |
|---------|------|-----|-----|-------|-------|-------|---------|
| | UP | RD | SM | | | | |
| | No | No | No | 78.79 | 80.55 | 61.35 | 70.77 |
| 5N 5K 2C | Yes | No | No | 79.25 | 83.07 | 62.27 | 72.34 |
| | Yes | Yes | No | 80.36 | 83.38 | 64.13 | 72.66 |
| | Yes | Yes | Yes | **85.35** | **87.63** | **68.33** | **76.93** |

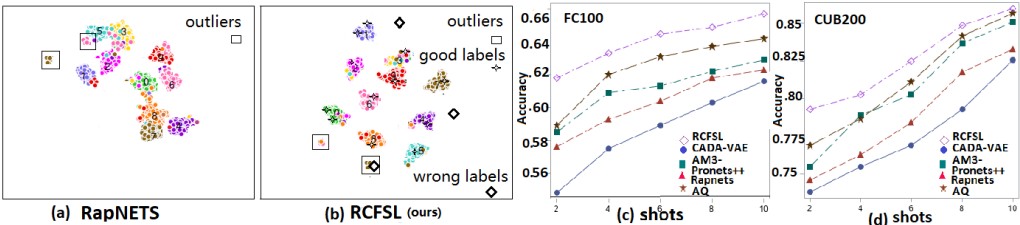

Figure 3: Comparison of the t-SNE visualization with two dimensional multimodal prototype space for the FewShot CIFAR-100 dataset using RapNetsLu et al. (2020b) and our RCFSL ($\beta$=0.2) on a 10-Way 8-shot semi-supervised learning (Fig 3.(a) and (b)), where each class includes 2 noisy labels and 2 outliers. The index number represents superclasses 0 to 9 in CIFAR-100. Each number locates on the median position of the corresponding vectors and the outliers are marked with squares. The embeddings from 10 distinct clusters using our method corresponds to true class labels, which validates the robustness of our method to label noise and outliers. Fig 3(c)- (d) illustrate the classification accuracy vs the number of shots for FC100 and CUB-200 datasets respectively.

remarkable in the fewer shots scenario, supporting the robustness of RCFSL. Similar performance evaluations are conducted on tieredImageNet dataset in Table 2 and RCFSL again is the top performer under various noise settings. Fig.3 demonstrates the comparison of the t-SNE visualization with two dimensional multimodal prototype space for the FewShot CIFAR-100 dataset using RapNetsLu et al. (2020b) and our RCFSL ($\beta$=0.2) on a 10-Way 8-shot semi-supervised learning (Fig 3.(a)- (b)), where it clearly shows our method has a better capability in terms of separating noisy labels and outliers in semi-supervised few-shot learning. We also investigate the classification accuracy of different methods versus the number of shots in Fig 3.(c)- (d) for FC100 and CUB200 datasets respectively with 25% noisy labels and outliers respectively, RCFSL yields considerable performance gains over competing methods especially with smaller number of shots.

**Ablation Study:** Described in the ablation study in Table 3, RCFSL w/o uncertainty prior excludes the uncertainty prior from the model. Therefore, the performance degradation suggests the importance of the proposed hierarchical structure for variational inference by placing the uncertainty prior on the infinite GMMs to counter the detrimental effects of outliers. Secondly, our method w/o robust divergence replaces $\beta$-divergence which places small weights on noisy labels and outliers with the regular KL-divergence. Moreover, RCFSL w/o cross modality relies on only image data for SFSL where multi-modality further improves the classification accuracy by at least 4%.

## 4 CONCLUSION

By integrating uncertainty prior of an infinite Gaussian mixture model into the heterogenous encoder and the robust lower bound based on $\beta$-divergence for variational inference, our RCFSL is capable of tackling the outliers and noisy labels simultaneously in semi-supervised few shot learning. Moreover, by integrating the robust semi-supervised GAN for feature generation, experimental evaluations on multiple benchmark datasets have demonstrate the superiority of the proposed method with large margins using multimodality data.

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

## A    DERIVATION OF INFLUENCE FUNCTION FOR LOWER BOUND OF RCFSL

Define $G(x)$ as a empirical cumulative distribution given by $\{x_i\}_{i=1}^n$ and denote the perturbed version of $G$ at $z$ as $G_{\varepsilon,z}(x) = (1-\varepsilon)G(x) + \Delta_z(x)$, where $\varepsilon$ is the contamination portion and $\Delta_z(x)$ is the point mass at $x$. Given a statistic $T$. the influence function (IF) is defined as Futami et al. (2018)

$$IF(z, T, G) = \frac{\partial T(G_{z,\varepsilon}(x))}{\partial \varepsilon} \mid_{\varepsilon=0} \tag{11}$$

Thus, the IF of the heterogenous variational autoencoder for $L_{RCVAE}$ is given by

$$
(\tfrac{\partial^2 L_{RCVAE}}{\partial H_u^2})^{-1} \tfrac{\partial}{\partial H_u} \mathbb{E}_{q(H_u)}[\mathrm{D}_{KL}[q(H_u|x,\mathbf{t})//p(H_u)]
$$
$$
+ N(\tfrac{\beta+1}{\beta} p(z|H_u)^\beta - \int p(x,\mathbf{t}|H_u)^{1+\beta} dx]
$$
$$
+ (\tfrac{\partial^2 L_{RCVAE}}{\partial H^2})^{-1} \tfrac{\partial}{\partial H} \mathbb{E}_{q(H)}[\mathrm{D}_{KL}[q(H|x,\mathbf{t})//p(H)]
$$
$$
+ N(\tfrac{\beta+1}{\beta} p(\tilde{y}|x,\mathbf{t},H)^\beta - \int p(\tilde{y}|x,\mathbf{t},H)^{1+\beta} d\tilde{y}] , \tag{12}
$$

It is straightforward to show that the above result is always bounded, namely RCFSL is robust to the compound noise (the outliers on the data $x$ and the labels $y$).

## B    VISUAL COMPARISON WITH ROBUST DIVERGENCE

Fig.1 demonstrates the comparison of the t-SNE visualization with two dimensional multimodal prototype space for the FewShot CIFAR-100 dataset using the same setting as our method except replacing robust divergence with KL-divergence and our RCFSL ($\beta$=0.2) on a 10-Way 8-shot semi-supervised learning, where it clearly shows the efficacy of the robust divergence by offering a stronger capability in separating noisy labels and outliers in semi-supervised few-shot learning.

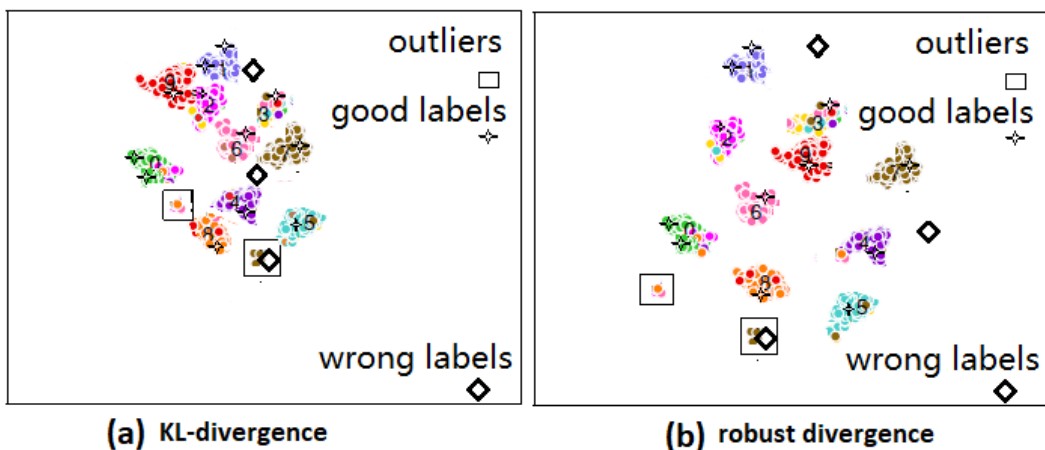

Figure 4: Comparison of the t-SNE visualization with two dimensional multimodal prototype space for the FewShot CIFAR-100 dataset using RCFSL with KL-divergence in variational inference and RCFSL with robust divergence ($\beta$=0.2) on a 10-Way 8-shot semi-supervised learning, where each class includes 2 noisy labels and 2 outliers. The index number represents superclasses 0 to 9 in CIFAR-100. Each number locates on the median position of the corresponding vectors and the outliers are marked with squares. The embeddings from 10 distinct clusters using our method corresponds to true class labels, which validates the robustness of our method to label noise and outliers.

Table 4: Comparison of the training time (hours) of RCFSL on Fewshot CIFAR-100 with several state-of-the-art approaches evaluated on a single Nvidia V100 GPU.

| CBPL-FSLXing et al. (2019a) | ReViSE-FSLTsai et al. (2017) | Feature transformTseng et al. (2020) |
|---|---|---|
| 6.3h | 7.6h | 5.9h |
| CADA-VAE-FSLSchönfeld et al. (2019) | AM3+TADAMXing et al. (2019b) | VHEraster-scan-GANZhang et al. (2020) |
| 7.2h | 6.3h | 8.3h |
| AQGoldblum et al. (2020) | RAPNetLu et al. (2020b) | RCFSL |
| 4.7h | 6.1h | 5.2h |

## C   DETAILS ON NETWORK STRUCTURE, TRAINING AND INFERENCE TIME COMPARISON

The discriminator of RCFSL following deep convolutional GAN is structured as follows: the first convolutional layer is of size $32 \times 32 \times 64$, the second convolutional layer is of dimension $16 \times 16 \times 128$, the third layer is $8 \times 8 \times 256$ and the last convolutional layer is of the dimension $4 \times 4 \times 512$ with the convolutional filter size $5 \times 5 \times 3$. Each convolutional layer is followed by a batch normalization layer and then a ReLU layer.

We analyze the training time (hours) and inference time (seconds) of our RCFSL to demonstrate its computational efficiency in Table 4 and Table 5. The training and the inference time of RCFSL is reported on Fewshot-CIFAR100 with several state-of-the-art approaches, evaluated on a single Nvidia V100 GPU. RCFSL is slower than AQ Goldblum et al. (2020), however is faster than the rest of competing methods for both training and inference time due to the efficient probabilistic graphical model and the effective novel lower bound and compact network architecture.

## D   DISCUSSION ON DIFFERENCES FROM OTHER MULTIMODAL VAES

Prior approaches Pandey & Dukkipati (2017) on multimodal VAEs mostly focus on cross-modal generation. In particular, given two modalities $x_1$ and $x_2$(e.g. image and text), these approaches

Table 5: Comparison of the inference time (seconds) of RCFSL on Fewshot CIFAR-100 with several state-of-the-art approaches evaluated on a single Nvidia V100 GPU.

| CBPL-FSLXing et al. (2019a) | ReViSE-FSLTsai et al. (2017) | Feature transformTseng et al. (2020) |
|---|---|---|
| 43s | 52s | 36s |
| CADA-VAE-FSLSchönfeld et al. (2019) | AM3+TADAMXing et al. (2019b) | VHEraster-scan-GANZhang et al. (2020) |
| 47s | 39s | 57s |
| AQGoldblum et al. (2020) | RAPNetLu et al. (2020b) | RCFSL |
| 17s | 37s | 23s |

learn the conditional generative model $p(x_1|x_2)$, where the conditioning modality $x_2$ and the generation modality $x_1$ are usually not interchangebale. Compared to Pandey & Dukkipati (2017), our heterogeneous robust VAE targets joint image-text modeling by explicitly modelling the joint distribution over latent variables from both image and textual data. Moreover, our multimodal VAEs integrates the uncertainty priors, denoising layers and robust variational lower bound to ensure the robustness in the presence of noisy labels and outliers. The work Suzuki et al. (2017) introduced joint multimodal VAE (JMVAE) which aims at learning shared representation with joint encoder $q_\psi(z|x_1, x_2)$. Distinct from Suzuki et al. (2017) which does not consider the robust modeling to tackle with outliers, our approach leverages infinite Gaussian mixture models equipped with uncertainty prior which not only characterizes the distributions for image-text modeling more accurately but also facilitates the learning of new unseen classes by inferring the number of classes automatically in few-shot learning. Recently, Zhang et al. (2020) utilized a VAE that encodes an image into a deterministic-upward–stochastic-downward ladder-structured latent representation, and then is applied to decode the corresponding text. While our work is relying on joint encoding and decoding schemes on image-text modelling and focusing on tackling the noisy-labels and outliers by enhancing the robustness of multimodal modeling using the various proposed techniques. More recently, Shi et al. (2019) leverages a mixture-of-experts multimodal variational autoencoder (MMVAE) to learn generative models on different sets of modalities, indicating the suitability of mixture models for learning from multi-modal data. Our work (RCFSL) advances MMVAE Shi et al. (2019) in three aspects: (1) We places uncertainty prior on the parameters of mixture models to avoid the collapsing of models in the presence of outliers and derived novel inference solution given the uncertainty prior (shown in our novel generative and inference models in Eqs (2)-(4)); (2) instead of using importance weighted autoencoder (IWAE) estimator Burda et al. (2015)Shi et al. (2019), RCFSL takes advantage of robust divergence to derive the robust evidence lower bound for unlabeled data and labeled data respectively which are unique and mathematically rigorously (e.g. Eq(7)), yielding superior performance for classification of noisy dataset in semi-supervised few-shot learning. (3) RCFSL integrates the denoising layers in both multimodal VAE and robust GAN to mitigate the influence of noisy labels, which further enhances the robustness of the model.

## E    EVALUATION ON SEMANTIC LABEL NOISE

In order to evaluate our algorithm under real-world noise (semantic label noise), RCFSL is further evaluated on Clothing 1M dataset with the network architecture Resnet-18. For Clothing1M dataset, it includes 1 million training images with 14 classes obtained from online shopping websites and labels are generated from surrounding texts. Since the data is collected from multiple online shopping websites and include many mislabelled samples, therefore the labels are contaminated by real-world noise (semantic label noise). The data is split such as $|C_{Base}| = 10$, $|C_{Novel}| = 4$. Table 6 demonstrates the comparison of classification accuracy using different learning algorithms on the Clothing1M datasets (real-world noise with 10% and 20% outliers respectively), where $n = 5$ images are selected from novel classes in order to perform few shot learning. As it can be seen from Table 6, our RCFSL continues to serve as the best performing method in the presence of semantic label noise compared to other competing methods under different noise statistics, which further confirms the outstanding learning capability of the proposed approach in robust semi-supervised few-shot learning. Moreover, Figure 5 illustrates exemplary images with compound noise detected

Table 6: Comparison of classification accuracy using different learning algorithms on the Clothing1M datasets (real-world noise with 10% and 20% outliers respectively) for 5 way 5 shot semi-supervised few shot learning on 1000 test episodes.

| Dataset | Clothing1M (10%) | Clothing1M (20%) |
|---|---|---|
| CBPL-FSLXing et al. (2019a) | 59.3 | 56.8 |
| ReViSE-FSLTsai et al. (2017) | 70.2 | 68.4 |
| Feature transformTseng et al. (2020) | 71.6 | 69.3 |
| AM3+TADAMXing et al. (2019b) | 71.8 | 70.5 |
| VHE-raster-scan-GANZhang et al. (2020) | 72.9 | 70.7 |
| AQGoldblum et al. (2020) | 72.6 | 70.1 |
| RGANKaneko et al. (2019) | 73.9 | 71.5 |
| RAPNetLu et al. (2020b) | 74.3 | 71.8 |
| RCFSL | **82.6** | **77.1** |

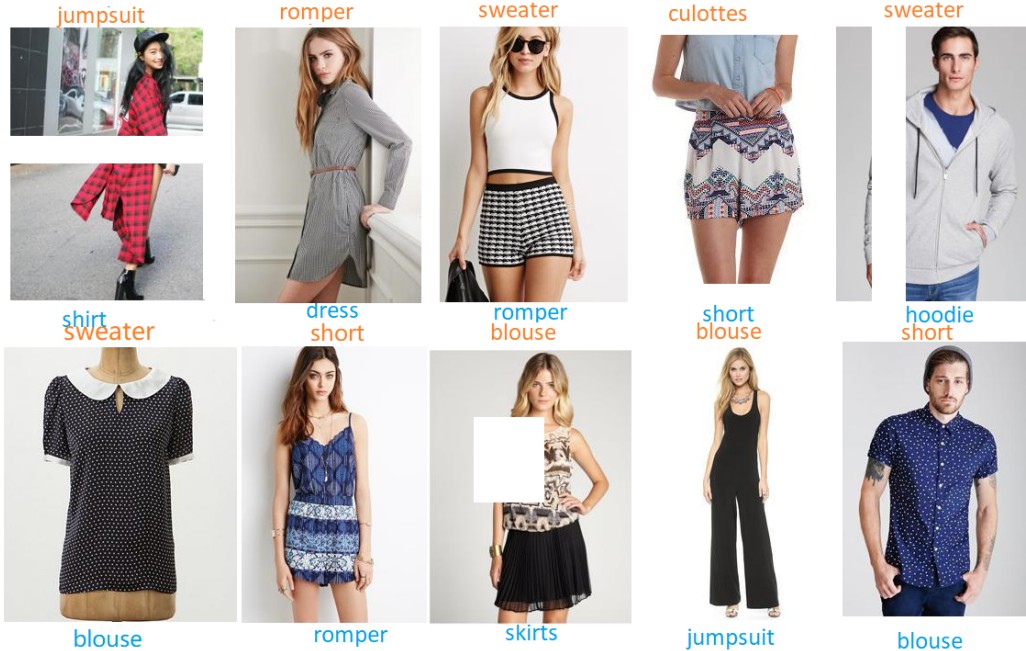

Figure 5: Exemplary images with compound noise detected by RCFSL from Clothing1M dataset, where the false labels are noted in orange and the true labels are noted in light blue.

by RCFSL from Clothing1M dataset, where the false labels are noted in orange and the true labels are noted in light blue.

## F EVALUATION ON OMNIGLOT DATASET

We further evaluate the proposed algorithm on Omniglot dataset. The Omniglot dataset includes 1623 different handwritten characters from 50 different alphabets where each of the 1623 characters was drawn online via Amazon's Mechanical Turk by 20 different people and each image is paired with stroke data, a sequences of spatial and temporal coordinates with in milliseconds. Table 7 demonstrates the comparison of classification accuracy using different learning algorithms on the Omniglot dataset (20% symmetric label noise with 10% outliers) for semi-supervised few shot learning on 1000 test episodes, where it can be seen from Table 7 that our RCFSL provides the best performance against other competing methods in the presence of noisy labels and outliers in semi-supervised few-shot learning.

Table 7: Comparison of classification accuracy using different learning algorithms on the Omniglot dataset (20% symmetric label noise with 10% outliers) for semi-supervised few shot learning on 1000 test episodes.

| Dataset | 5 way 5 shot | 5 way 10 shot |
|---|---|---|
| CBPL-FSLXing et al. (2019a) | 95.2 | 97.8 |
| ReViSE-FSLTsai et al. (2017) | 95.6 | 98.1 |
| Feature transformTseng et al. (2020) | 96.3 | 98.3 |
| AM3+TADAMXing et al. (2019b) | 96.2 | 98.2 |
| VHE-raster-scan-GANZhang et al. (2020) | 96.5 | 98.6 |
| AQGoldblum et al. (2020) | 96.8 | 98.7 |
| RGANKaneko et al. (2019) | 97.1 | 98.9 |
| RAPNetLu et al. (2020b) | 97.3 | 99.1 |
| RCFSL | **97.9** | **99.7** |

Table 8: Unsupervised test log-likelihood using different learning algorithms on the permutation invariant MNIST dataset (with 20% outliers) the normalizing flows VAE (VAE+NF), importance weighted auto-encoder (IWAE), variational Gaussian pro-cess VAE (VAE+VGP), Ladder VAE (LVAE) with FT denoting the finetuning procedure Sønderby et al. (2016) and auxiliary deep generative models Maaløe et al. (2017) and our method ($\beta$=0.2), where $L$ represents the number of stochastic latent layers $z_1, \ldots, z_L$ and $IW$ characterizes the importance weighted samples during training.

| Method | $-\log p(x)$ |
|---|---|
| VAE+NFMiyato et al. (2015), L=1 | -89.35 |
| IWAE, L=1, IW=1 Burda et al. (2015) | -90.26 |
| IWAE, L=1, IW=50 Burda et al. (2015) | -88.36 |
| IWAE, L=2, IW=1 Burda et al. (2015) | -89.71 |
| IWAE, L=2, IW=50 Burda et al. (2015) | -86.43 |
| VAE+VGP, L=2 Tran et al. (2015) | -85.79 |
| LVAE, L=5, IW=1 Sønderby et al. (2016) | -85.08 |
| ADGM, L=1, IW=1 Maaløe et al. (2017) | -84.67 |
| ADGM, L=1, IW=2 Maaløe et al. (2017) | -84.34 |
| RCFSL+ KL divergence, L=1, IW=2 | -83.76 |
| RCFSL (robust divergence), L=1, IW=2 | **-81.35** |

## G    COMPARISON OF LOG-LIKELIHOOD WITH ROBUST DIVERGENCE

We report the roust lower bound for the unlabeled data with 5000 importance weighted samples where the similar setting as Rasmus et al. (2015) with warm up, batch normalization and 1 Monte Carlo and IW sample for training. The percentage of outliers is set to be 20%. Table 8 demonstrates the log-likelihood scores for the permutation invariant MNIST dataset. The results shown in Table 8 indicates the the proposed method has strong expressive power by performing better than other methods in terms of log-likelihood due to the utilization of the robust divergence in the inference, especially by comparing with the same RCFSL method but with KL-divergence.

## H    ANALYSIS FOR DIFFERENT TERMS IN THE OPTIMIZATION FUNCTION

In the above optimization cost function of RCFSL in the equation (10), the first term represents the supervised and unsupervised loss from the discriminator, the second and the third terms tend to min-imize the approximation error by "regularizing" $q(H|x, \mathbf{t}, \tilde{y})$ and $q(H_u|x, \mathbf{t})$ to the prior $p(H)$ and $p(H_u)$ for labeled data and unlabeled data respectively. The fourth term characterizes the generator loss from the robust semi-supervised GAN (RSGAN). The fifth to the eighth terms are employed to enhance the robustness in the presence of data outliers with $\beta$-divergence for the ELBOs on the

labeled data and unlabeled data respectively, which is shown to be insensitive to the small contamination of the data.

