# OpenReview forum: "Robust Cross-Modal Semi-supervised Few Shot Learning"
_ICLR.cc/2022/Conference — ICLR 2022 Submitted_

### Official Review · Reviewer_JD5c · 2021-10-31

**Correctness:** 3
**Technical Novelty And Significance:** 2
**Empirical Novelty And Significance:** 3
**Recommendation:** 6
**Confidence:** 2

**Main Review:**

# Strengths
1. The idea of introducing Bayesian deep learning to the semi-supervised few-shot learning problem sounds reasonable.
2. The provided experimental results show the effectiveness of the proposed method to some extent.

# Weaknesses
1. Why could the introduced uncertainty prior mitigate the impact of corrupted labels and outliers?  How to ensure the relationship between the uncertainty prior and the noisy data? It is suggested to give a more intuitive explanation.
2. The authors claim that their robust variational lower bound can infer network parameters more accurately. However, it is hard to see the advantages of the robust variational lower bound intuitively from the paper.
3. It seems that there are multiple loss terms in the proposed objective function, such as Eq. (10). How does the author determine their respective weights during optimization? Do these weight parameters have much influence on the results?
4. It is suggested to give more implementation details, such as the network structure, experimental configuration, training and inference time.
5. The writing of this paper needs to be further polished. For example, "the parameters of infinite Gaussian mixture model" -> "the parameters of an infinite Gaussian mixture model"? "the performance gains shows" -> "the performance gains show"?


**Summary Of The Paper:**

This paper focuses on semi-supervised few-shot learning with multi-modality data. The authors introduce an uncertainty prior of an infinite Gaussian model, integrating multi-modality information from image and text data into a heterogenous variational autoencoder. Meanwhile, a new variational lower bound is derived for the inference of parameters. In addition, a GAN is developed for the data sparsity in few shot learning. Experimental results demonstrate the effectiveness of the proposed method to some extent.

**Summary Of The Review:**

Overall, this paper provides a new perspective on the cross-modal semi-supervised few-shot learning.

---

> ### Author Response · Authors · 2021-11-12
> **Response to Reviewer JD5C (Part I)**
>
> First of all, we would like to thank the reviewer for the effort and valuable comments as these comments are really helpful. We are encouraged by the positive comments on "new prospective", "reasonable approach and the effectiveness of proposed method in experimental evaluations". We then address each of the reviewer's questions as follows:
>
> (1)	“Why could the introduced uncertainty prior mitigate the impact of corrupted labels and outliers? How to ensure the relationship between the uncertainty prior and the noisy data? It is suggested to give a more intuitive explanation.
>
> To address this concern, the revised paper further adds one more reference with more clear statements in both of introduction and Section 2 ("our method"). The first work to place a Gaussian prior on top of the mean for Gaussian component is from S. Hou and A. Galata, “Robust Estimation of Gaussian Mixtures from Noisy Input Data” in CVPR 2008: “As the level of uncertainties on the input data increases, maximum likelihood algorithms tend to return mixtures with some very narrow components which are collapsing onto a point or a hyperplane”, therefore, it is very desirable to place priors on the parameters of the mixture components which constrains the shape of the components to prevent them from collapsing (such as Fig.2 described in the reference (Hou & Galata,2008) more intuitively). However,  (Hou & Galata, 2008) is mainly designed for clustering in unsupervised learning and therefore it does not consider the more complicated and interesting problem of robust cross-modal semi-supervised few shot learning.
>
> Typically, in the latest paper, we clarified that "Firstly, motivated by (Hou & Galata, 2008) , it places the uncertainty prior of the parameters of infinite Gaussian mixture distribution of image data to avoid mixture components collapsing onto a point or a hyperplane due to outliers". Moreover, “Inspired by the fact that the student-t distribution is more robust to the outliers than Gaussian distribution by constraining the shapes of the mixture components from collapsing (Hou & Galata, 2008) , we propose to place uncertainty priors (e.g. Gaussian priors) on parameters of infinite Gaussian mixture models to characterize the influence of outliers on image data and constrain the shape of the components to prevent them from collapsing" in page 3 in Section 2.
>
> (2)	“The authors claim that their robust variational lower bound can infer network parameters more accurately. However, it is hard to see the advantages of the robust variational lower bound intuitively from the paper.”
>
> The revised paper has conducted two additional analysis to further show the advantages of robust variational lower bound intuitively.
> (a) for the visual comparison of the 2-dimensional tsne embedding using KL-divergence and robust divergence in Fig.4 in the appendix Section B which provides an intuitive way to qualitatively evaluate the effectiveness of inferring network parameters more accurately with the proposed robust variational lower bound. Meanwhile, in the table 3, the ablation study on robust divergence also highlight the advantages of the robust variational lower bound in terms of classification accuracy.
>
> (b) For numerical comparison, the roust lower bound is reported for unlabeled data with 5000 importance weighted samples  for training in Section G in the appendix. The percentage of outliers is set to be 20\%. Table 8 demonstrates the log-likelihood scores for the permutation invariant MNIST dataset. The results shown in Table 8 indicates the the proposed method has strong expressive power by performing better than other methods in terms of higher log-likelihood due to the utilization of the robust divergence in the inference, especially by comparing with the same RCFSL method but with KL-divergence.
>
> (3)	“It seems that there are multiple loss terms in the proposed objective function, such as Eq. (10). How does the author determine their respective weights during optimization? Do these weight parameters have much influence on the results?”
>
> The revised paper has clarified that there are two weight parameters in the proposed objective functions: $\lambda_1$ and $\beta$. Here $\lambda_1$ is utilized to control the trade-off between unlabeled data and labeled data and is set to be the ratio of labeled samples and unlabeled samples, which is similar to [Maaloe, 2017] and turns out to be the best selections for $\lambda_1$. We further clarify that the best selections of $\lambda_1$ can improve the overall classification accuracy 0.5\% compared to random selection. Also as stated in the Section of “Implementation Details”, $\beta$ varies between 0.1 and 0.4 and the best performance is reported. We also report that by tuning $\beta$, the classification accuracy can be further boosted by 0.6\% on average across all datasets. Moreover, a detailed analysis for loss function on each term in the loss function in Section H in the appendix.

---

> > ### Author Response · Authors · 2021-11-21
> > **Response to Reviewer JD5C (Part II)**
> >
> > (4) “It is suggested to give more implementation details, such as the network structure, experimental configuration, training and inference time.
> >
> > For network structure and experimental configuration, we clarified that in the revised paper that "we parameterize the heterogeneous VAE with three sets of 5 by 5 fully convolutional, ReLU and pooling layers followed by two fully connected hidden layers where each pair of layers contains the hidden units as dim(h) = 1000. We further set the dimension of the auxiliary variable a and the latent variable z to be 200 respectively. For the CUB dataset, we leverage Resnet-18 as backbone and incorporate the uncertainty prior for the input, the noise transition model along with the auxiliary variables as our encoder. ... For the configurations of discriminator in GAN, we follow the same configuration of DC-GAN and follow the similar discriminator Mirza & Osindero (2014) for DC-GAN." We further included more details about network structure on the discriminator in Section C in the appendix "The discriminator of RCFSL following deep convolutional GAN is structured as follows: the first convolutional layer is of size $32\times 32\times 64$, the second convolutional layer is of dimension $16 \times 16 \times 128$, the third convolutional layer is $8 \times 8 \times 256$ and the last convolutional layer is of the dimension $4\times 4 \times 512$ with the convolutional filter size $5\times 5\times 3$. Each convolutional layer is followed by a batch normalization layer and then a ReLU layer.".
> >
> > The revised paper has also included the comparison of the training and inference time for the proposed method with respect to the state of the art approaches in the appendix Section C in Table 4 and Table 5. Evaluated on a single Nvidia V100 GPU on Fewshot-CIFAR 100 dataset, our RCFSL is slower than AQ[Goldblum et al 2020], however is faster than the rest of competing methods due to the efficient probabilistic graphical model and the effective novel lower bound and compact network architecture.
> >
> > (5) "The writing of this paper needs to be further polished. For example, "the parameters of infinite Gaussian mixture model" -> "the parameters of an infinite Gaussian mixture model"? "the performance gains shows" -> "the performance gains show"? "
> >
> > We have carefully proofread and polished the writing of the paper in the revised version as the reviewer pointed out in the revised version. Thanks.

---

### Official Review · Reviewer_RfHj · 2021-11-01

**Correctness:** 4
**Technical Novelty And Significance:** 2
**Empirical Novelty And Significance:** 3
**Recommendation:** 8
**Confidence:** 3

**Main Review:**

Paper strengths:

+ The proposed approach relies on existing methodologies that are combined in a new way.

+ The problem is clearly described and the paper is well-written.

+ The approach delivers good performance in comparison to the related approaches.

Paper weaknesses:

- The paper focuses on image-text multi-modal learning. In the introduction, this is can be only implicitly inferred after reading the introduction. A clear problem statement would be necessary.

- Novelty: It is not clear in which sense is the variational autoencoder (VAE) heterogeneous. The term is actually not defined in the context of the problem. Moreover, relying on text and images to train the VAE is not unusual. It should be clarified what is heterogeneous and why it is novel.

- Although the paper is well-written, it relies on several approaches which have to be described. The current structure of the paper does not help to understand the complete approach. For instance, the explanation of the uncertainty prior is not clear. Moreover, the method section lacks an overall description of the approach prior to describing the details. Overall, the method section could be improved to help the reader easier follow the paper's idea.

- Symmetric and asymmetric noise are not often encountered in real-world problems. It would be more interesting to see semantic label noise being considered.


Improvements:

- Omniglot could be considered in the evaluations too. It is also a standard benchmark.

**Summary Of The Paper:**

The paper presents a cross-modal semi-supervised few-shot learning approach for image classification. The idea is to train variational auto-encoder (VAE) with both image and text data for learning a feature representation. Then features are extracted from a test sample and assigned to the class of the closest prototype train sample. In addition, a generative adversarial objective is employed for learning the latent code of the VAE. Since the proposed idea is meant for learning from noisy labels, there is an uncertainty prior in the image features as part of the infinite Gaussian mixture distribution. The approach is evaluated on standard few-shot learning benchmarks which are modified to account for the noise labels. The results are promising compared to the prior work.

**Summary Of The Review:**

Overall, the paper addresses the problem of noisy labels next to cross-modal semi-supervised few-shot learning. This problem formulation makes the proposed work different from the existing approaches. Moreover, the proposed formulation could be considered novel regardless of combining existing approaches. The main issue of the paper is the difficulty to follow the proposed approach. One needs to read several times the paper for understanding the structure of the proposed method. This is a major point that needs improvement. Finally, considering semantic label noise would be the whole idea more plausible for real-world setups.

Post-rebuttal comment:

The rebuttal has helped to improve the paper. My concerns have been addressed.

---

> ### Author Response · Authors · 2021-11-11
> **Response to Reviewer RfHj (Part I)**
>
> We thank the reviewer for the valuable comments and detailed suggestions. We appreciate the positive comments from the reviewer on "novel ideas", "new approaches with the proposed formulation", "well-written paper" as well as "good performance in comparison to the related approaches".
>
> (1)	“The paper focuses on image-text multi-modal learning. In the introduction, this is can be only implicitly inferred after reading the introduction. A clear problem statement would be necessary.
>
> We agree with the reviewer’s suggestions and have explicitly highlight the image-text multi-modal learning in the revised paper in the abstract and the introduction. For instance, in the abstract of revised paper, it is mentioned that “we propose to employ a robust image-text multi-modal semi-supervised few-shot learning (RCFSL) based on Bayesian deep learning.” In the 2nd paragraph of the  introduction, it is stated that “Incorporating image-text multi-modal learning into the framework by training on image-text pairs provides an efficient tool to inject the diversity to the generation process …” and “by integrating a deep
> generative heterogenous model that generalizes well to multi-modality (e.g. image-text modeling) in order to counter noisy labels and outliers …” Thanks for the comments and indeed we feel that these revisions make the problem statement more clear.
>
> (2)	“It is not clear in which sense is the variational autoencoder (VAE) heterogeneous.
> The term is actually not defined in the context of the problem. Moreover, relying on text and images to train the VAE is not unusual. It should be clarified what is heterogeneous and why it is novel”
>
> In the revised paper, we emphasize that the definition of the heterogeneous VAE comes from its capability of consuming both continuous (image) and discrete (text) features as input. The revised paper further clarifies that the novelty of the proposed heterogeneous VAE is that “to the best of our knowledge, this is the first time that a robust heterogeneous VAE model has been applied which naturally integrates noisy images and text together which cohesively fuse continuous and discrete multi-modality features” highlighted in the Section 2 “The heterogeneous mixture model”. Namely, the unique probabilistic modeling for image-text data in a robust manner including uncertainty priors on infinite GMM differentiates our approach from traditional methods and results in significantly improved performance.
>
> (3)	“For instance, the explanation of the uncertainty prior is not clear.”
>
> The revised paper has clarified that “Inspired by the fact that the student-t distribution is more robust to the outliers than Gaussian
> distribution by constraining the shapes of the mixture components from collapsing, we propose to place uncertainty priors (e.g. Gaussian priors) on parameters of infinite Gaussian mixture models to characterize the influence of outliers on
> image data” in the first paragraph of the Section 2. Typically, the Gaussian prior is placed on the mean of each Gaussian component to characterize the uncertainty and outliers from the input.
>
> (4)  “Moreover, the method section lacks an overall description of the approach prior to describing the details. Overall, the method section could be improved to help the reader easier follow the paper's idea.”
>
> Thanks for the comments. The revised paper has provided an overall description of the approach prior to describing the details right after the subtitle of the method section in page 3. Typically, it is stated that “Our
> approach is to focus on first constructing a semi-supervised robust heterogeneous variational autoencoder leveraging a mixture model to encode both image and text data in few shot learning insensitive to both noisy labels and outliers. Subsequently, a novel robust variational lower bound is derived to facilitate the inference of network parameters relying on $\beta$-divergence for both labeled and unlabeled data. Finally, a robust generative adversairial network is integrated with denoising layers to strengthen the denoising performance togehter with the end-to-end optimization to generate additional visual features to alleviate the sparsity in the
> semi-supervised few shot learning.” We have further improved the write-up the method section to help the reader follow the idea of the paper easily in the revised paper.

---

> > ### Comment · Reviewer_RfHj · 2021-11-18
> > **More clarifications on the response**
> >
> > It would be great a provide a few more details on the uncertainty prior (concern of all reviews): why placing a Gaussian (on the top of the mean for each Gaussian component) is a measure of uncertainty? The argument is not well-motivated even in the updated version. Moreover, it is not still clear what is the difference of the proposed VAE (in terms of formulation) from existing multi-modal VAE implementation that jointly learn from text and image data.

---

> > > ### Author Response · Authors · 2021-11-18
> > > **Re: More clarifications on the response**
> > >
> > > “It would be great a provide a few more details on the uncertainty prior (concern of all reviews): why placing a Gaussian (on the top of the mean for each Gaussian component) is a measure of uncertainty? The argument is not well-motivated even in the updated version”
> > >
> > > We have submitted a latest revised paper to address this concern further by adding one more reference with more clear statements in both of introduction and Section 2 ("our method").
> > > The first work to place a Gaussian prior on top of the mean for Gaussian component is from S. Hou and A. Galata, “Robust Estimation of Gaussian Mixtures from Noisy Input Data” in CVPR 2008. As mentioned in this reference, “As the level of uncertainties on the input data increases, maximum likelihood algorithms tend to return mixtures with some very narrow components which are collapsing onto a point or a hyperplane”, therefore, it is very desirable to place priors on the parameters of the mixture components which constrains the shape of the components to prevent them from collapsing (such as Fig.2 described in the reference (Hou & Galata,2008) more intuitively). However, the method described in (Hou & Galata, 2008) is mainly designed for clustering in an unsupervised learning setting and therefore it does not take into account of the more complicated and interesting problem of robust cross-modal semi-supervised few shot learning as we addressed in this paper.
> > >
> > > Typically, in the latest paper, in the introduction, we clarified that "Firstly, motivated by (Hou & Galata, 2008) , it places the
> > > uncertainty prior of the parameters of infinite Gaussian mixture distribution of image data to avoid mixture components collapsing onto a point or a hyperplane due to outliers". Moroover, it is stated “Inspired by the fact that the student-t
> > > distribution is more robust to the outliers than Gaussian distribution by constraining the shapes of the mixture components from collapsing (Hou & Galata, 2008) , we propose to place uncertainty priors (e.g. Gaussian priors) on parameters of infinite
> > > Gaussian mixture models to characterize the influence of outliers on image data and constrain the shape of the components to prevent them from collapsing" in page 3 in Section 2.  Please let us know if there are further questions and comments along this line and we are happy to reply. Thanks.
> > >
> > > “Moreover, it is not still clear what is the difference of the proposed VAE (in terms of formulation) from existing multi-modal VAE implementation that jointly learn from text and image data.”
> > >
> > > We have clarified the difference of the proposed VAE (in terms of formulation) from existing multi-model VAE implementation that jointly learn from text and image data in the Section D. Typically, in the latest revised paper, it is mentioned that “ ...Compared to [Pandey & Dukkipati (2017)], our
> > > heterogeneous robust VAE targets joint image-text modeling by explicitly modelling the joint distribution over latent variables from both image and textual data. Moreover, our multimodal VAEs
> > > integrates the uncertainty priors, denoising layers and robust variational lower bound to ensure the
> > > robustness in the presence of noisy labels and outliers. The work Suzuki et al. (2017) introduced
> > > joint multimodal VAE (JMVAE) which aims at learning shared representation with joint encoder
> > > $q (z_j|x_1; x_2)$. Distinct from Suzuki et al. (2017) which does not consider the robust modeling to
> > > tackle with outliers, our approach leverages infinite Gaussian mixture models equipped with uncertainty
> > > prior which not only characterizes the distributions for image-text modeling more accurately
> > > but also facilates the learning of new unseen classes by inferring the number of classes automatically
> > > in few-shot learning. More recently, Shi et al. (2019) leverages a mixture-of-experts multimodal
> > > variational autoencoder (MMVAE) to learn generative models on different sets of modalities, indicating
> > > the suitability of mixture models for learning from multi-modal data. Our work (RCFSL)
> > > advances MMVAE Shi et al. (2019) in three aspects:
> > >  (1) We places uncertainty prior on the parameters
> > > of mixture models to avoid the collapsing of models in the presence of outliers and derived novel
> > > inference solution given the uncertaity prior (shown in our novel generative and inference models
> > > in Eqs (2)-(4));
> > >
> > >  (2) instead of using importance weighted autoencoder (IWAE) estimator Shi et al.
> > > (2019), RCFSL takes advantage of robust divergence to derive the robust evidence lower bound for
> > > unlabeled data and labeled data respectively which are unique and mathematically rigorously (e.g.
> > > Eq(7)), yielding superior performance for classification of noisy dataset in semi-supervised few-shot
> > > learning.
> > >
> > > (3) RCFSL integrates the denoising layers in both multimodal VAE and robust GAN to
> > > mitigate the influence of noisy labels, which further enhances the robustness of the model. "
> > >
> > > Please feel free to let us know if there are further questions and comments. Thanks !

---

> > > > ### Comment · Reviewer_RfHj · 2021-11-26
> > > > **Final Score**
> > > >
> > > > Thank you for the feedback. My concerns are addressed and the increased my score to Accept.

---

> > ### Author Response · Authors · 2021-11-19
> > **Response to Reviewer RfHj (Part II)**
> >
> > (5) "Symmetric and asymmetric noise are not often encountered in real-world problems. It would be more interesting to see semantic label noise being considered. "
> >
> > First of all, asymmetric noise is really designed to cope with real-world problems by replacing the correct labels with the most similar classes. Secondly, we agree with the reviewer that it will be even better to consider the semantic label noise case. Therefore,
> > the revised paper has included the evaluation with semantic label noise in Section E in the appendix. In order to evaluate our algorithm under real-world noise (semantic label noise), RCFSL is further evaluated on Clothing 1M dataset with the network architecture Resnet-18.
> > For Clothing1M dataset, it includes 1 million training images obtained from online shopping websites and labels are generated from surrounding texts where the labels are contaminated by real-world noise. The data is
> > split such as $|C_{Base}|=150$, $|C_{Novel}|=50$. Table 6 demonstrates the comparison of classification accuracy using different
> > learning algorithms on the Clothing1M datasets (real-world noise with 10\% and 20\% outliers respectively), where $n=5$ images are selected from novel classes in order to perform few shot learning. As shown in Table 6, RCFSL continues to serve as the best performer under semantic label noise with different noise statistics compared to other competing methods for 5 way 5 shot semi-supervised few shot learning on 1000 test episodes, which further confirms the superiority of RCFSL.
> >
> > (6) "Omniglot could be considered in the evaluations too. It is also a standard benchmark. "
> >
> > We have further conducted experiments on the Omniglot dataset and compared the classification performances with other competing methods in Section F in the appendix in the revised paper. it can be seen from Table 7 that our RCFSL provides the best performance against other competing methods in the presence of noisy labels and outliers in semi-supervised few-shot learning as expected.

---

### Official Review · Reviewer_SEhY · 2021-11-08

**Correctness:** 3
**Technical Novelty And Significance:** 2
**Empirical Novelty And Significance:** 2
**Recommendation:** 5
**Confidence:** 2

**Main Review:**

This paper provides enough details of their method, and shows performance improvement over some existing methods. However, I have major concerns about the problem formulation, the motivation of the method, and the experiments.

This paper aims to address four problems at the same time: (1) few-shot learning, (2) multimodal FS learning, (3) semi-supervised FS learning, and (4) FS learning under data noise. It is not clear what are the unique challenges posed by each problem, and why this paper combines all the problems together to create a new and complex problem. There need to be more intuitive justifications on the importance of the problem studied.

The method is also quite complex and not well-motivated. It is unclear what challenge doe each component address. It is also difficult for me to understand what value does the proposed mixture model offer that is specific to the problem studied.

The experiment setting is confusing to me. Could the author explain in more details about the (1) multimodal data, (2) semi-supervised setting, and (3) dataset noise. It seems to me that all the baseline results are reproduced on the new setting (please correct me if I'm wrong), because previous methods are not targeted to this problem. In such cases, the improvements over baselines are expected. The authors either need to adapt the baseline, or provide more justifications on why the baselines are strong enough.

**Summary Of The Paper:**

This paper proposes to address the complex problem of cross-modal semi-supervised few-shot learning with noisy data. A rather complex approach is proposed with combines Bayesian mixture mode, VAE, GAN, and prototypical learning.

**Summary Of The Review:**

This paper has limitations in the problem formulation, the motivation of the method, and the experiments. In my opinion, it needs major improvement.

---

> ### Author Response · Authors · 2021-11-11
> **Response to Reviewer SEhY (Part I)**
>
> We thank the reviewer for the valuable comments and feedback. We respond to the reviewer’s comments as follows:
>
> (i)	“This paper aims to address four problems at the same time: (1) few-shot learning, (2) multimodal FS learning, (3) semi-supervised FS learning, and (4) FS learning under data noise. It is not clear what are the unique challenges posed by each problem, and why this paper combines all the problems together to create a new and complex problem.”
>
> First of all, both R2 and R3 have provided positive comments on this point. "Overall, this paper provides a new perspective on the cross-modal semi-supervised few-shot learning."(by R3). "The problem is clearly described and the paper is well-written" (by R2).
>
> Subsequenlty, we further clarify that the formulations of the problem are focused on tackling the uncertainty in learning with low data problems, where the low data problem means that there are little labeled data (images) for each class. In this situation, (1) few-shot learning serves as a powerful tool for learning tasks (including unseen classes) when there are rare labeled data in each category. To mitigate the scarcity of labeled data, here we leverage two commonly used solutions in FS learning (2) multimodal FS learning to harness rich information from text with image in the Bayesian deep learning (3) semi-supervised FS learning to exploit the information from massive unlabeled data. By introducing (2) and (3), in the revised version, we have highlighted in the introduction that in page 2 “now the question then is: how to design a Bayesian deep learning which counters the noisy labels and outliers jointly in the multimodal semi-supervised few-shot learning. Accordingly, this paper tackles this challenging problem in robust cross-modal few-shot learning by integrating a deep generative heterogenous model that generalizes well to multi-modality (e.g. image-text modeling) in order to counter noisy labels and outliers.”
>
> (ii)	“The method is also quite complex and not well-motivated. It is unclear what challenge doe each component address.”
>
> First of all, Please note that both of R2 and R3 have provided positive comments on this point. As R2 mentioned that "The problem is clearly described and the paper is well-written". R3 also commented that "The idea of introducing Bayesian deep learning to the semi-supervised few-shot learning problem sounds reasonable".
>
> Next, we further clarify that the challenges that each component addresses as follows: (a) the uncertainty prior is designed to mitigate the influence of outliers as it models the mean of each Gaussian components as Gaussian distribution to characterize the variation of the data. (b)  The denoising layers applied in both robust VAE and robust GAN have effectively reduce the influences from noisy labels by applying the noise transition matrices so that the posterior distributions of latent variables are conditioned on corrected labels. (c)  The robust divergence is helpful for reducing the detrimental effects of both noisy labels and outliers by assigning small or zero weights on them.  The revised paper has further highlighted and validated the contributions of each component in ablation study quantitatively.  Moreover, the visual illustration of comparison using robust divergence added in Section B further validates the effectiveness of robust divergence. Thanks.
>
>
> (iii)	“It is also difficult for me to understand what value does the proposed mixture model offer that is specific to the problem studied.”
>
> We clarify that the advantages of using infinite Gaussian mixture model (iGMM) for few shot learning are three-fold:
>  (1) Inspired from [Allen, 2019] ,  the strength of combining deep representation learning with Bayesian nonparametrics is that representing each class by a set of clusters can improve the classification accuracy. However, the work  [Allen, 2019]  neither focuses on the realistic case with both noisy labels and outliers, nor does it consider the multimodal learning based on image and text data jointly in few-shot learning. (2) the capability of iGMM in learning new classes from unseen samples in testing dataset also plays a critical role in the learning process by inferring the number of classes automatically. (3) By placing the uncertainty prior on top of the parameters of iGMM (e.g. the mean of each Gaussian component), the hierarchical representation of  the proposed probabilistic mixture model further offers robustness in the presence of outliers which is the key to the overall performance gains.
>
> The revised paper also emphasized the value of the proposed mixture model to the problem studied as" image features are modeled as
> infinite Gaussian mixture distribution [Allen et al, 2019] to characterize the samples from the unseen classes by computing
> the posterior distribution for unrepresented clusters more accurately
> and inferring the number of classes automatically".

---

> > ### Author Response · Authors · 2021-11-19
> > **Response to Reviewer SEhY (Part II)**
> >
> > (iv) “The experiment setting is confusing to me. Could the author explain in more details about the (1) multimodal data, (2) semi-supervised setting, and (3) dataset noise.”
> >
> > Please note both of R2 and R3 provide positive comments on the experimental sections such as "The provided experimental results show the effectiveness of the proposed method" by R3 and "The approach delivers good performance in comparison to the related approaches" by R2.
> > We further clarify that these details are described clearly in the Section 3 “Dataset and Competing Methods”.
> >   (1) The multimodal data consists of both image and text data. For image data, " It includes mini-ImageNet(mIN), tieredImageNet (tIN), Fewshot-CIFAR100 (FC100) and Caltech-UCSD Birds 200-2011 (CUB). The miniImageNet dataset is a subset of ILSCRS-12 which includes 100 classes where each class contains 600 RGB images with size 84 by 84. We split the datasets by using 64 classes
> > for training, 16 classes for validation and 20 classes for testing where 16 classes are utilized to monitor the model’s generalization performance. The CUB Bird dataset contains 11788 images of 200 different bird species, where the data is split equally in training and testing data.”  For text data “10 short textual descriptions per image are provided by Reed et al. (2016). GloVe Pennington
> > et al. (2014) is utilized to extract the word embeddings for the category labels, where the embeddings are trained with large unsupervised text corpora.”
> >
> > (2) The semi-supervised setting is conducted as follows: “Namely, for each category, 30 training image and 30 testing images are included, where in the training dataset, 10% images are labeled images and the rest of the images are unlabeled.
> >
> > (3) the dataset noise includes noisy labels and outliers. Typically, for noisy labels “we evaluate the two types of label noise including symmetric noise and asymmetric noise. For symmetric noise, we inject the label noise by randomly flipping the labels
> > of the labeled data into a different label in the novel classes. For asymmetric noise, the corrupted
> > labels are generated by replacing the correct labels with their most similar classes (using nearest
> > neighbor measurement).” For outliers, “Outliers are mimicked by including samples from data distributions far away from the training data distributions, namely out-of-distribution (OOD) samples.” Therefore, we feel that the experimental setting is stated clearly. Please let us know if there are further questions. Thanks.
> >
> > (V) “It seems to me that all the baseline results are reproduced on the new setting (please correct me if I'm wrong), because previous methods are not targeted to this problem. In such cases, the improvements over baselines are expected. The authors either need to adapt the baseline, or provide more justifications on why the baselines are strong enough. "
> >
> > We would like to clarify that the baseline methods are selected and set up in the similar way as the reference by Xing et al “Adaptive cross modal few-shot learning” (Neurips 2019), where three families of methods are compared as baselines and STOA approaches including (1) Uni-modality few-shot learning baselines. (2) Cross-modality baseline extended to metric-based FSL work. (3) Robust few-shot learning methods. Therefore, we believe that we have followed a standard way to produce the baseline results. The revised paper has highlighted this point in the experimental comparison session as well by stating "Similarly as [Xing et al, 2019b], three families of methods are evaluated" in the section 3 page 7.
> >
> > Namely for the first family, the base line methods are evaluated with uni-modality relying on the same semi-supervised noisy datasets. For the second family, the baseline approaches introduced here are already adapted to the new setting under the same input noisy datasets which are strong enough for comparison. Therefore, we feel that it is a good justification that the baselines are strong enough for comparison. Thanks.

---

> > > ### Comment · Reviewer_SEhY · 2021-11-26
> > > **Thanks for the response**
> > >
> > > The detailed response is very much appreciated, which clears up some of my doubts. However, I still believe it would be better if the proposed method can be evaluated on each individual low data problem, where it can be compared to the sota in each field. Therefore, I would remain my original score.

---

> > > > ### Author Response · Authors · 2021-11-26
> > > > **Re: Thanks for the response for Reviewer SEhY**
> > > >
> > > > "The detailed response is very much appreciated, which clears up some of my doubts. However, I still believe it would be better if the proposed method can be evaluated on each individual low data problem, where it can be compared to the sota in each field. Therefore, I would remain my original score."
> > > >
> > > > We thank the reviewer (R1) for the additional comments and are glad to hear that the response clears up some of the reviewer’s doubts. Firstly, we further clarify that for each of individual low data problem including semi-supervised learning, multi-modal learning and few-shot learning is essentially a very broad field. Like other reviewers (R2 and R3) mentioned, the main contribution of this paper is to “provide a new perspective on cross-modal  semi-supervised few-shot learning using Bayesian deep learning (by R3)”.  As R2 mentioned, “The approach is evaluated on standard few-shot learning benchmarks which are modified to account for the noise labels. The results are promising compared to the prior work”. As R3 commented "The provided experimental results show the effectiveness of the proposed method ".
> > > >
> > > > Subsequently, we emphasize that the current experimental results with comparison to three family methods are already sufficient enough to demonstrate the superiority of our method over the state-of-the-art approaches. They include (1) Uni-modality few-shot learning baselines; (2) Cross-modality baseline extended to metric-based FSL work; (3) Robust few-shot learning methods which already cover the SOTA methods in this field similarly as Xing et al “Adaptive cross modal few-shot learning” (Neurips 2019).
> > > >
> > > > Moreover, we clarify that we have also compared our method to each individual low data problem including semi-supervised learning and multi-modal methods and as expected:
> > > >
> > > > (1) the traditional semi-supervised learning methods (without few-shot learning) usually donot work well in few-shot learning settings (in the sense that the classification accuracy is much lower than our method). That is because frequently there are only limited images (e.g. less than 5) in each class and there are also unseen classes in few-shot learning.
> > > >
> > > > (2) the traditional multi-modal or cross-modal learning methods (without few-shot learning) donot have the capability to handle noisy labels and outliers and are not able to learn well with limited number of labeled data. Thus the performance of those methods typically collapses in the presence of complicated noise, where that is essentially the strengths and the novelty of our method.
> > > >
> > > > We have already mentioned the above points in the introduction and will add more notes in the camera-ready version of the paper. Thanks.

---

### Author Response · Authors · 2021-11-20
**New revised paper has been submitted (with highlighted revisions as follows)**

Dear Reviewers:

Thank you very much for providing the careful reviews and constructive comments on the paper. We are grateful for positive comments including "addresses the problem of noisy labels next to cross-modal semi-supervised few-shot learning. This problem formulation makes the proposed work different from the existing approaches" (R2) , " a new perspective on the cross-modal semi-supervised few-shot learning" (R3), "reasonable idea of introducing Bayesian deep learning to semi-supervised few shot learning" (R3), "clearly described paper and well-written paper" (R2), "delivers good performance in comparison to the related approaches" (R2) and "the effectiveness of the proposed method in experimental results" (R3).  Based on the reviews, we have submitted a latest revised version which incorporate the modifications to address the shared concerns. Below we first summarize the highlighted revisions (including for the shared concerns) and then address specific comments from each reviewer in separate responses.

(1) The revised paper has explained the intuition of using uncertainty priors more clearly for R2 and R3 in both of the introduction and the beginning of section 2 (to avoid the mixture components in Gaussian mixture model collapsing into a point or a hyperplane due to outliers). Meanwhile, we also include a related reference to further provide more details in order to address the concern.

(2) The revised paper has been enhanced with multiple sections in the appendix including:

a. Section B (Visual Comparison on Robust Divergence) and Section G (Comparison on Log-likelihood on Robust Divergence) for R1 and R3 to better demonstrate the advantages of robust divergences in addition to the ablation study in the main body paper.

b. Section C (Details on Network Structure, Training and Inference Time Comparison) for R3 on details of implementation.

c. Section D (Discussion of the difference on multimodal VAEs) for R2 in terms of formulation to highlight the novelty of our robust multimodal VAE.

d. Section E (Semantic Label Noise) by evaluating on Clothing 1M dataset and Section F (evaluation on Omniglot dataset) for R2 with additional experiments and analysis.

e. A detailed analysis of the meaning of each term in the loss function in the equation (10) (for R3).

(3) The revised paper has also included an overall description of the approach prior to describing the details at the beginning of Section 2 and highlighted the image-text modeling explicitly in both of the abstract and the introduction (for R2).

(4) The revised paper includes more related references and highlight for better motivation of the usage for infinite Gaussian mixture model and the setup of the comparison of baseline methods in experiments. (for R1)

We greatly appreciate all reviewers' suggestions. Furthermore, we hope that our paper updates and responses have addressed reviewers' questions and concerns. Please let us know if there are further questions.

---

### Decision · Program_Chairs · 2022-01-20

**Decision:**

Reject

**Comment:**

This paper aims to address the problem of cross-modal semi-supervised few-shot learning with noisy data, and proposed a robust cross-modal semi-supervised few-shot learning (RCFSL) based on Bayesian deep learning. The approach combines several existing techniques for tackling a new problem in a non-trivial approach. Empirical results demonstrate the effectiveness of the proposed method to some extent.

While the proposed integrated complex approach seems to be novel in the proposed unique setting, there are some major concerns from the reviewers. One concern is about the lack of clear justification on technical contributions for the proposed methodology in the complex settings. In particular, it lacks of comprehensive ablation studies for analyzing and understanding the source of gains by the proposed complex method, and the baselines in the experiments also do not look strong enough. In addition, many aspects of the paper writing and presentation are not satisfied (e.g., the math formulation in Section 2 is densely presented making it difficult to follow).

Overall, this is a borderline case, where the paper did contribute a new method for the interesting cross-modal semi-supervised few-shot learning task, but some major concerns on the weaknesses remain at its current form. Therefore, it cannot be recommended for acceptance. Nonetheless, I hope authors can improve the paper by fully addressing these issues and hope to see it accepted in the near future.